# MagicPIG: LSH Sampling for Efficient LLM Generation

**Zhuoming Chen[1], Ranajoy Sadhukhan[1], Zihao Ye[2], Yang Zhou[1], Jianyu Zhang[34], Niklas Nolte[4]**
**Yuandong Tian[4], Matthijs Douze[4], Leon Bottou[4], Zhihao Jia[1], Beidi Chen[1]**
[1]Carnegie Mellon University [2]University of Washington [3]New York University [4]Meta AI
{zhuominc,rsadhukh,yangzhuo6,zhihaoj2,beidic}@andrew.cmu.edu
zhye@cs.washington.edu
{jianyuzhang,nolte,yuandong,matthijs,leonb}@meta.com

## Abstract

Large language models (LLMs) with long context windows have gained significant attention. However, the KV cache, stored to avoid re-computation, becomes a bottleneck. Various dynamic sparse or TopK-based attention approximation methods have been proposed to leverage the common insight that attention is sparse. In this paper, we first show that TopK attention itself suffers from quality degradation in certain downstream tasks because attention is not always as sparse as expected. Rather than selecting the keys and values with the highest attention scores, sampling with theoretical guarantees can provide a better estimation for attention output. To make the sampling-based approximation practical in LLM generation, we propose MagicPIG, a heterogeneous system based on Locality Sensitive Hashing (LSH). MagicPIG significantly reduces the workload of attention computation while preserving high accuracy for diverse tasks. MagicPIG stores the LSH hash tables and runs the attention computation on the CPU, which allows it to serve longer contexts and larger batch sizes with high approximation accuracy. MagicPIG can improve decoding throughput by up to $5\times$ across various GPU hardware and achieve 54ms decoding latency on a single RTX 4090 for Llama-3.1-8B-Instruct model with a context of 96k tokens.

## 1 Introduction

Large language models (LLMs) with long context windows, such as GPT (Achiam et al., 2023), Llama (Dubey et al., 2024), and Gemini (Team et al., 2023), have gained significant attention for their ability to enhance applications like chatbots (Chiang et al., 2024), search engines (Wang et al., 2024), and video analysis (Cheng et al., 2024). However, serving long-context LLMs is highly challenging due to the unique bottleneck in auto-regressive generation—the key-value (KV) cache, which stores intermediate attention keys and values to avoid re-computation (Pope et al., 2022; Zhang et al., 2023b). Specifically, the KV cache grows linearly with both the batch size and sequence length, occupying substantial GPU memory and increasing decoding time. Moreover, the KV cache makes LLM generation extremely memory-bound, leading to underutilization of GPU computational power. For instance, an NVIDIA A100-40GB GPU can only handle a single request for Llama with a 128k context length, with nearly half of the decoding time spent accessing the KV cache, and poor GPU utilization (He & Zhai, 2024).

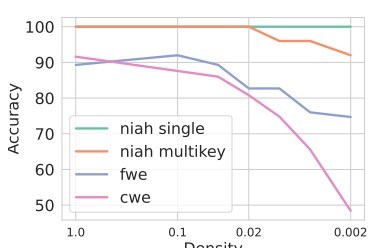

**Figure 1:** While TopK attention performs well on information retrieval tasks (niah) where the useful information reduces to a few words, it degrades severely in harder aggregated tasks like word extraction (cwe, fwe). x-axis: proportion of attention keys used for TopK attention.

Leveraging the common insight that attention is naturally sparse, dynamic sparse or TopK-based approximation has been extensively studied (Tang et al., 2024; Singhania et al., 2024; Zhang et al., 2024; Liu et al., 2024a), but three major challenges prevent a wide adoption in LLM serving systems. (1) **Quality Degradation.** They usually propose various strategies to approximate a subset of KV cache that yields the highest attention scores. However, TopK attention itself is a biased attention

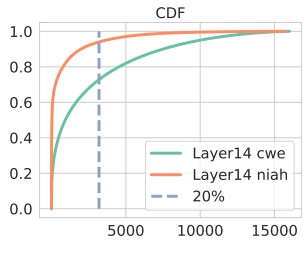 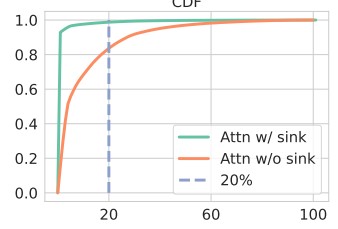 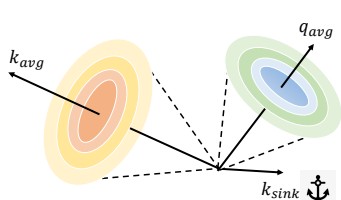

**(a)** Long tailed phenomena     **(b)** Attention sink reshapes sparsity     **(c)** Geometry of attention

**Figure 2: Left:** Examples of long tailed distribution in LLM. The x-axis is the fraction (or number of tokens) used in the $\mathrm{TopK}$, a.k.a. the *sampling budget*. **Mid:** Sink tokens make attention score look sparser. **Right:** The geometry of attention. The key states of attention sink $k_0$ is almost opposite to other tokens and its orientation is surprisingly invariant of input tokens. Query states lie close to $k_0$, thus forming attention sink and Figure 2b. $k$ usually lie in a narrow cone that is far away from $q$. In certain heads, this geometry will result a long tailed distribution of attention score as well as the difficulty to search for the $\mathrm{TopK}$ keys.

approximation and lacks theoretical guarantees. Figure 1 shows that even exact $\mathrm{TopK}$ attention results significantly degrade the accuracy of certain downstream tasks. (2) **High Overhead.** There is a large overhead to identify $\mathrm{TopK}$ attention, which becomes the bottleneck rather than the attention computation. For example, as studied in Liu et al. (2024a), naively applying a search algorithms like IVF (Douze et al., 2024) requires to access over $30\%$ key states to obtain the exact $\mathrm{TopK}$, showing an unsatisfying trade-off between search accuracy and cost. (3) **No Memory Saving.** Although saving KV cache loading time, they cannot reduce the total memory occupied by the KV cache, which limits the maximum context and batch sizes when VRAM is scarce.

An ideal sparse attention approximation approach should (1) preserve full accuracy for a diverse set of downstream tasks with guarantees, (2) involve low-cost overhead for KV cache selection, and (3) save GPU memory. The following observations together with the performance drop shown in Figure 1 suggest that to achieve such demanding requirements, we need to go beyond $\mathrm{TopK}$ attention:

- *Attention is not always sparse.* Contradictory to previous belief (Zhang et al., 2023b; 2024; Tang et al., 2024; Liu et al., 2024a), we observe that attention is not always sparse, especially for tasks which leverage the full context. As shown in Figure 2a, in some layers, attention distribution can be very long-tailed, *i.e.*, the $\mathrm{Top}20\%$ attention can only cover $70\%$ of the total attention scores.

- *Seemingly high sparsity is usually a consequence of an attention sink.* Most of the attention scores concentrate on initial tokens (attention sink phenomenon) (Xiao et al., 2023), making the distribution look sparser. However, as shown in Figure 2b, attention scores are distributed more uniformly among tokens except for the sink. According to the geometrical interpretation of sink, keys, and queries shown in Figure 2c, the attention sink, which we found surprisingly almost static regardless of the input token, is just for imposing sparsity on the attention distribution.

- *It is hard to find $\mathrm{TopK}$ attention.* Figure 2c also shows why searching for the Top-K keys is intrinsically costly. The keys and queries usually lie within two narrow cones with nearly opposite orientations, except for the attention sink. This significant mismatch between query and data distributions causes nearest-neighbor search methods to perform poorly.

These limitations of $\mathrm{TopK}$ attention requires rethinking the sparse attention approximation. Rather than only using the keys and values with highest scores, leveraging information on the distribution can make the estimation more accurate. We approach this as as bias correction problem in sampling. Unbiased and efficient sampling has been long studied in biology (Lukacs, 2009), sociology (Chen et al., 2018) as well as machine learning (Backurs et al., 2019; Chen et al., 2019; Zandieh et al., 2023), with theoretical guarantees.

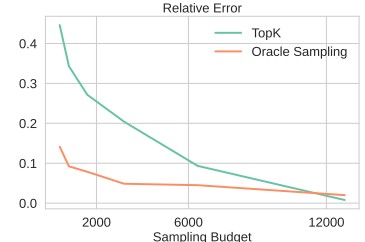

**Figure 3:** $\mathrm{TopK}$ v.s. Sampling, 16k total context

Figure 3 shows that sampling values according to their corresponding attention score (we call this *oracle sampling*) achieves a much lower (up to $4\times$) estimation error than the naive $\mathrm{TopK}$ selection. Deploying sampling estimation in attention is promising, but three challenges remain. First, how a reduction of the attention error can make a difference in downstream performance is

unclear (Backurs et al., 2019; 2018). Second, modeling the attention scores distribution is necessary for efficient sampling, but inferring the distribution parameters requires expensive computations. Third, fully leveraging the resources of modern hardware, GPU and CPU, with a theoretically efficient algorithm is non-trivial.

In this paper, we propose Magic samPlIng for Generation (MAGICPIG), which leverages Locality sensitive hashing (LSH) sampling for efficient LLM generation. LSH is employed for sampling to approximate the attention scores distribution and estimate attention output. By computing hash functions on GPU and conducting sampling on CPU, MAGICPIG can allow massive hash tables and hash functions compared to prior work (Kitaev et al., 2020; Chen et al., 2021), which are of vital importance for accurate estimation (Backurs et al., 2018). Following the practice of Aminabadi et al. (2022); He & Zhai (2024), we offload the KV cache computation, which is memory bound, to CPU to allow a larger batch or longer context. Specifically,

- In Section 3, we analyze the failures of TopK attention. Moreover, we study sampling based attention estimation assuming an oracle for the key distribution (**Oracle Sampling Estimation**) in detail and empirically demonstrate that it is consistently more effective both for in distribution estimation and on downstream tasks.
- In Sections 4.1 to 4.3, we present a sampling algorithm to approximate oracle sampling for attention estimation based on locality sensitive hashing and the intuition and motivation from statistic perspectives. To our best knowledge, MAGICPIG is the first to leverage LSH sampling in self attention in decoder-only LLM generation.
- In Section 4.4, we present our system design to efficiently offload attention computation on CPU, breaking the memory limit of GPU for serving larger batch or longer contexts. We also overcome the new challenges of computation and memory size raised by our sampling algorithm to support a larger scale of hashing tables beyond prior work (Chen et al., 2021; Kitaev et al., 2020).

In Section 5, we show the empirical evaluation results of the performance of MAGICPIG, demonstrating the accuracy and efficiency. While maintaining high accuracy for diverse tasks, MAGICPIG can improve serving throughput by $1.5 \sim 5\times$ (A100, L20, RTX 4090) and can achieve 54ms decoding latency on a single RTX 4090 for Llama-3.1-8B-Instruct (Dubey et al., 2024) with 96K context. More importantly, we show that MAGICPIG already outperforms TopK attention in the two aggregation tasks in Figure 1, suggesting that sampling indeed goes beyond TopK attention.

## 2 BACKGROUND

In this section, we formulate the targeted attention estimation problem and related works.

### 2.1 PROBLEM FORMULATION

In LLM decoding phase, self attention part calculates an weighted-average of previous values by

$$o = \text{Softmax}(\frac{qK^T}{\sqrt{d}})V = wV \quad q \in \mathbb{R}^{1\times d} \quad K, V \in \mathbb{R}^{n\times d} \quad w \in \mathbb{R}^{1\times n} \tag{1}$$

where $d$ is the head dimension and $n$ is the context size. $K = [k_1, k_2, ..., k_n], V = [v_1, v_2, ..., v_n], k_i, v_i \in \mathbb{R}^{1\times d}$ is KV cache. Normalized attention weight $w = \text{Softmax}(\frac{qK^T}{\sqrt{d}}) \in \mathbb{R}^{1\times n}$ is also called attention (score) distribution. Our target is to find sampling matrix $\Pi \in \mathbb{R}^{n\times m}$ and diagonal matrix $D \in \mathbb{R}^{m\times m}$ which minimize

$$\delta = ||wV - w\Pi D\Pi^T V|| \tag{2}$$

where $m \ll n$ is computation budget. For TopK attention, suppose $w_{r_1} > ... > w_{r_m} > ... > w_{r_n}$, then

$$\Pi_{i,j} = \begin{cases} 1, & \text{if } i = r_j, \\ 0, & \text{otherwise.} \end{cases} \quad D_{ii} = \frac{1}{\sum_{i=1}^{m} w_{r_i}} \tag{3}$$

### 2.2 RELATED WORKS

**Efficient Attention.** Attention approximation has been long studied. Reformer (Kitaev et al., 2020), KDEformer (Zandieh et al., 2023), HyperAttention (Han et al., 2023) and ScatterBrain (Chen et al., 2021) tackle the problem via locality sensitive hashing. These methods work in training and encoding (e.g., BigGAN (Brock et al., 2019)). Theoretically, the error bounds and minimal workload required are continuously improved (Brand et al., 2023; Alman & Song, 2023) but not proven to be practical for wall-clock acceleration in LLM decoding. Flash-attention (Dao et al., 2022b; Dao, 2023; Dao et al.,

2022a), flash-decoding (Ye et al., 2024; Hong et al., 2024) and SlimAttention (He et al., 2024)accelerate attention operator by maximizing hardware utilization, which is orthogonal to our approach.

**Locality sensitive hashing.** Locality sensitive hashing (LSH) (Backurs et al., 2019; 2018) is a family of hashing functions which assigns the same hash codes for similar inputs with higher probability than others (Chen et al., 2020b; Jafari et al., 2021). LSH uses two hyper-parameters, $(K, L)$. $L$ hash tables are independently built. Each hash table has its own function $H$ which projects a high dimension vector to an integer by concatenating $K$ random independent hash functions. In the sampling process, all vectors which share hash codes in at least one hash tables with query will be collected. **SimHash** (Charikar, 2002) is the LSH family based on cosine similarity. For a vector $x \in \mathbb{R}^d$, SimHash generates a random hyperplane $w$ and returns $\text{Sign}(w^T x)$. Vectors share the same sign if and only if the random projection is not in-between them. For a random projection, all angles are equally likely, thus the probability that two vectors $x$, $y$ share the same sign for is $p = 1 - \frac{\theta}{\pi}$, where $\theta = \arccos \frac{xy^T}{||x|| \cdot ||y||}$. If we have $L$ hash tables each with $K$ random hash functions, the probability of $y$ to be retrieved by query $x$ is $1 - (1 - p^K)^L$.

**KV Cache reduction.** To get rid of memory bound introduced by KV cache thus enabling a larger batch size or serving a longer prompt, many methods are proposed to reduce the volume of KV cache. For example, $H_2O$ (Zhang et al., 2023b), SnapKV (Li et al., 2024) and Keyformer (Adnan et al., 2024) calculates heuristics during prefilling phase to decide which tokens to preserve for decoding phase. Quest (Tang et al., 2024) and Loki (Singhania et al., 2024) do not evict KV cache but apply dynamic sparsity to reduce KV Cache loading at inference time. StreamingLLM (Xiao et al., 2023) and InfLLM Xiao et al. (2024) reduce attention computation and support extremely long context by context window extrapolation. Methods like KIVI (Liu et al., 2024b) and QServe (Lin et al., 2024) reduce the size of KV Cache by quantization.

## 3 RETHINKING ATTENTION SPARSITY

In this section, we examine TopK attention, which is the theoretical upper bound of prior search-based algorithms including both static methods (Zhang et al., 2023b; Li et al., 2024) and dynamic methods (Tang et al., 2024; Singhania et al., 2024; Mao et al., 2024). We show that TopK is *sub-optimal* and present another attention approximation based on sampling and estimation with an oracle, that improves the accuracy and/or the computation cost.

### 3.1 ACHILLES' HEEL OF TOPK ATTENTION

As it is defined, TopK attention only computes the weighted average on elements with highest attention scores. To quantify its performance, the *computation budget* of TopK attention, is defined as the number of selected tokens, i.e. the K of TopK. Searching-based sparse attention algorithms, like (Tang et al., 2024; Singhania et al., 2024; Wu et al., 2024) are approximations for TopK attention by replacing the true TopK keys with the ones found by approximate searching algorithms.

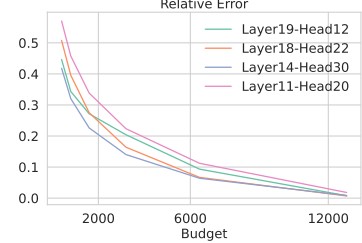

**Figure 4:** TopK estimation error for a KV-cache of 16k tokens.

However, we do find a significant performance degradation in downstream tasks caused by TopK attention as shown in Figure 1. Although TopK attention preserves accuracy for retrieval tasks that only require a minimal subset of the context (needle-in-a-haystack, niah, single/multikey (Hsieh et al., 2024)), it severely degrades for aggregation tasks that leverage the full context (common word extraction, cwe, and frequent word extraction, fwe (Hsieh et al., 2024)). Intuitively, the information is distributed more broadly for aggregation tasks, which results in less peaky attention score distribution.

TopK attention is *biased* and *inaccurate* especially when the distribution of attention scores is long tailed and the computation budget or density (i.e. $K$) is limited. Long tail phenomena do occur in LLMs across all layers as presented in Figure 2a. Top20% tokens can only cover $70 \sim 80\%$ attention scores, leaving a large proportion of keys and values not considered, which is translated into a non-negligible ($15 \sim 20\%$) estimation error in Figure 4. More observations are presented in Appendix B.

### 3.2 ESTIMATE ATTENTION WITH SAMPLING

Existing TopK attention mechanisms ignore tokens in the KV cache with low attention scores, which introduces a bias, since the ignored tokens sum up to a large proportion of attention scores (Figure 2a). As a result, TopK attention achieves suboptimal performance for long-context tasks, such as information

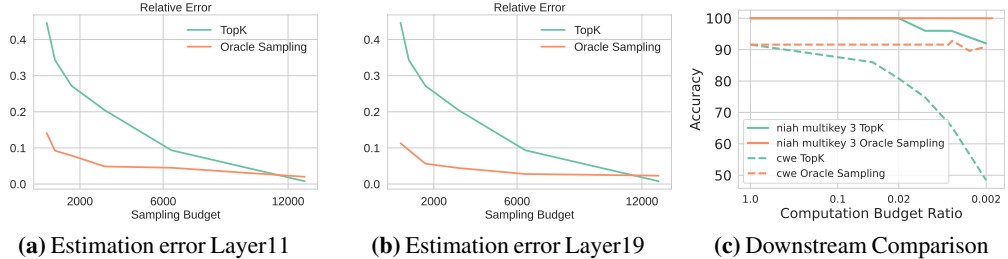

**Figure 5: Left and Middle:** Oracle sampling estimation can significantly reduce numerical error compared to $\mathrm{Top}K$ attention. The evaluated context size is 16k. The $x$-axis is *sampling budget* for oracle sampling and *computation budget* for $\mathrm{Top}K$ attention. Notice that the estimation error of $\mathrm{Top}K$ attention will cross oracle sampling after a certain large budget (12k in figures). The reason is that oracle sampling will repetitively sample the same subset of tokens with a high probability while $\mathrm{Top}K$ will not. Theorem 3.3 further explains this. **Right:** Downstream comparison for oracle sampling estimation and $\mathrm{Top}K$ attention. The $x$-axis for both methods is *computation budget ratio*, i.e. the fraction of selected/sampled tokens.

aggregation (Figure 1). Increasing the computation budget for TopK attention does help reduce the estimation error (Figure 4) since it will involve more elements in computing, however, the following question is posed:

*Can we improve the estimation quality with low computational budgets?*

Inspired by *mark and recapture* (Lukacs, 2009; Owen, 2013; Lohr, 2021; Chen et al., 2018), we show in the following that attention output can be estimated with sampling. Using notations from Section 2.1 we can re-write attention output $o$ as the expectation of $v_i, 1 \leq i \leq n$ from distribution $w$, i.e. $o = \mathbb{E}_{i \sim w}(v_i)$, which can be estimated by the following method.

**Definition 3.1** (Oracle Sampling Estimation). Given a sampling budget $\mathcal{B}$ and normalized attention score $w$, $\mathcal{B}$ elements are sampled independently from $w$ (i.e. $i_1, i_2, ..., i_{\mathcal{B}} \overset{iid}{\sim} w$). Then the attention output is estimated as

$$\bar{o} = \frac{1}{\mathcal{B}} \sum_{j=1}^{\mathcal{B}} v_{i_j} \tag{4}$$

This is not the lowest variance estimator but has a better downstream performance (see Appendix C). We call it "oracle" because it assumes that the exact attention vector $w$ is known, which is not true for sparse attention approximations.

**Theorem 3.2.** *Oracle sampling estimation is unbiased and the trace of covariance is monotonically decreasing with $\mathcal{B}$.*

This theorem (proved in Appendix A) theoretically guarantees a low estimation error of oracle sampling. We also present an empirical comparison between oracle sampling estimation and $\mathrm{Top}K$ attention in Figures 5a and 5b. In summary, oracle sampling estimation can reduce relative error by up to $4\times$.

Note that the sampling budget $\mathcal{B}$ is not the actual computation cost for oracle sampling estimation: duplicate $X_i$ need to be computed/loaded only once, so $\bar{o}$ can be computed by

$$\bar{o} = \sum_{i \in \mathcal{S}} \frac{f_i}{\mathcal{B}} v_i \quad S = \mathrm{Unique}(\{i_{1 \leq i \leq \mathcal{B}}\}) \tag{5}$$

where $f_i$ is the number of duplicates of $X_i$. Intuitively, if $w$ has an peaked distribution (e.g. $w_i > 99\%$), then almost all samples in $\{i_1, ..., i_{\mathcal{B}}\}$ are identical to $i$. The actual computation cost of oracle sampling estimation is $|S|$, the number of *unique* samples, which we bound in the following:

**Theorem 3.3.** *The expected computation budget ($\mathbb{E}(|S|)$) has an upper bound of $1 + \mathcal{B}\epsilon$, where $\epsilon = 1 - \max_i w_i$.*

This theroem (proved in Appendix A) shows that the computation cost of oracle sampling is usually far less than the sampling budget. In Figure 5c, we present the downstream accuracy comparison between oracle sampling estimation and $\mathrm{Top}K$ attention. The former preserves high accuracy for both tasks, even with very small computation cost ($0.002\%$ out of 16k context, which is approximately 32).

## 4 MAGICPIG

Section 3.2 demonstrates the potential of sampling-based estimation. In Sections 4.1 and 4.2, we present how we arrives at Locality sensitive hashing to unleash this potential from a statistical perspective. In Section 4.3, we show the practical algorithm. Finally, in Section 4.4, we demonstrate our system co-design for accurate and efficient LLM decoding through GPU-CPU collaboration.

Note that most of the derivations in this section might be classical and can even be found in textbooks, but our goal is to leverage them to motivate MAGICPIG design and precisely demonstrate the power of a rigorously sound algorithm with system co-design in deep generative models.

### 4.1 SELF-NORMALIZED IMPORTANCE SAMPLING FOR ATTENTION ESTIMATION

Oracle sampling estimation cannot go beyond $2\times$ wall clock speed up because obtaining distribution $w$ requires full computation of all $qk_i^T$ and thereby only saving the $wV$ computation.

Fortunately, importance sampling (Kloek & Van Dijk, 1978; Owen, 2013; Lohr, 2021) allows us to perform estimation for unknown distribution $w$ by sampling from a proposed distribution $u$. In our problem setting, the normalization factor of $w$, i.e. $Z = \sum_{i=1}^{n} \exp \frac{qk_i^T}{\sqrt{d}}$ is also unknown because computing it requires evaluating all $qk_i^T$. However, we do have access to unnormalized weights $\widetilde{w_i} = e^{\frac{qk_i^T}{\sqrt{d}}}$ for sampled indices $i$. Hence, by employing a variant of importance sampling, **self-normalized importance sampling** (Owen, 2013), we sample indices $i_1, i_2, ..., i_{\mathcal{B}}$ from a proposed distribution $u$ and the resulting estimator is

$$X^{\mathrm{IS}} = \frac{1}{\widetilde{Z}} \sum_{j=1}^{\mathcal{B}} \frac{\widetilde{w_{i_j}}}{u_{i_j}} v_{i_j} \quad \text{where} \quad \widetilde{Z} = \sum_{j=1}^{\mathcal{B}} \frac{\widetilde{w_{i_j}}}{u_{i_j}} \tag{6}$$

which has a very nice property for accurately estimating attention output that $\mathbb{P}[\lim_{k \to \infty} X^{\mathrm{IS}} = o] = 1$. Its variance[1] is related to the distribution $u$, and can be approximated by

$$\widetilde{\mathrm{Var}}(X^{\mathrm{IS}}) = \frac{1}{\mathcal{B}} \mathbb{E}_{i \sim u}\left[\frac{w_i^2}{u_i^2}(v_i - o)^2\right] = \frac{1}{\mathcal{B}Z^2} \mathbb{E}_{i \sim u}\left[\frac{\widetilde{w_i}^2}{u_i^2}(v_i - o)^2\right] \tag{7}$$

To minimize the variance, $u$ should satisfy $u \propto \widetilde{w_i}|v_i - o|$ (Hesterberg, 2003). The variance will be high if $u_i$ and $\widetilde{w_i}|v_i - o|$ assign a high probability mass to different regions of the sample space or have different modes. Therefore, the challenge is compute a distribution $u$ aligned with $\widetilde{w_i}|v_i - o|$ without accessing too many $\widetilde{w_i}$. Besides, Equation (6) requires that sampling probability $u$ can be computed and $u_i > 0$, which is not satisfied by many deterministic approximations like TopK.

### 4.2 VARIANCE REDUCTION WITH LSH

We decompose $\widetilde{w_i}|v_i - o| = \exp(\frac{qk_i^T}{\sqrt{d}} + \log|v_i - o|)$. We observe emprically (Figure 10 in the appendix) that $\log|v_i - o|$ does not fluctuate significantly compared to $\frac{qk_i^T}{\sqrt{d}}$. Hence, we simplify the requirement of $u$ to share the same peaks with $qk_i^T$. By the following transformation,

$$r = \max_{1 \le i \le n} |k_i| \quad \bar{q} = [q, 0] \quad \bar{k_i} = [k_i, \sqrt{r^2 - |k_i|^2}] \tag{8}$$

we further transfer inner product $qk_i^T$ to cosine similarity between $\bar{q}$ and $\bar{k_i}$ (which is a common practice in Maximum Inner Product Search (Shrivastava & Li, 2014)).

Inspired by prior work (Spring & Shrivastava, 2017; Chen et al., 2020a), we leverage Locality sensitive hashing-based sampling for this estimation problem. Specifically, leveraging a hash function $h$ in the LSH family that preserves cosine similarity such as SimHash (Sadowski, 2007), we can sample from probability distribution $u_i = \mathbb{P}[h(q) = h(k_i)]$ which is monotonic to $\cos \frac{qk_i^T}{|q| \cdot |k_i|}$.

### 4.3 ALGORITHM IMPLEMENTATION

To make this estimation practical, MAGICPIG is implemented by the following specific design.

**Estimator approximation.** Self-normalized important sampling Equation (6) requires $i_1, i_2, ..., i_k$ iid sampled but the probabilities provided by hashing are not normalized. Hence we adapt our estimator:

---

[1] We assume head dimension $d = 1$ here for simplicity. Higher dimension has similar formulations and analysis by replacing variance with trace of covariance.

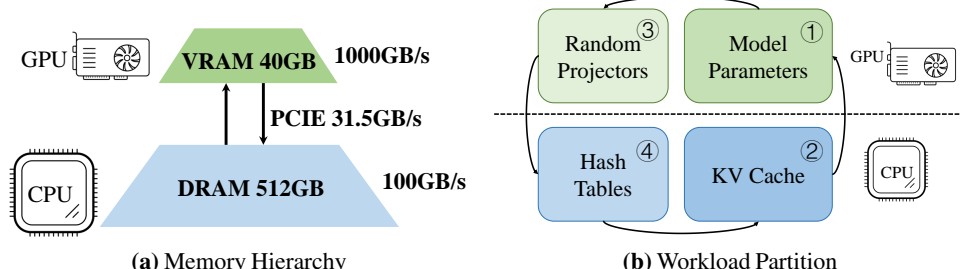

**(a)** Memory Hierarchy        **(b)** Workload Partition

**Figure 6: Left:** Memory hierarchy of hardware. GPU VRAM has high bandwidth but is limited. CPU DRAM is sufficient but is relatively slow. The limited bandwidth of PCIE forbids large-scale data transferring. **Right:** Workload partition of MAGICPIG. Linear projections and hash function computation (by random projection) are done on GPU, while sampling with hash tables and attention are done CPU. The execution order is ①③④②.

After obtaining $S$ with probability $u$, MAGICPIG computes

$$X = \frac{\sum_{i=1}^{n}\frac{\widetilde{w_i}}{u_i}v_i\mathbf{1}_{i\in S}}{\sum_{i=1}^{n}\frac{\widetilde{w_i}}{u_i}\mathbf{1}_{i\in S}} = \frac{\sum_{i\in S}\frac{\widetilde{w_i}}{u_i}v_i}{\sum_{i\in S}\frac{\widetilde{w_i}}{u_i}} \qquad (9)$$

**Hash function selection.** MAGICPIG leverages **SimHash** (Sadowski, 2007), that draws with $K \times L$ random vectors. For each of the $L$ hash tables, the $q$ and $k_i$s vectors are projected on $K$ directions and only the sign of the projection is kept, which yields a $K$-bit hash value. Key $k_i$ is sampled only if there exist at least **two** hash tables where $k_i$ shares the hash value with $q$. The corresponding probability is

$$u_i = \mathbb{P}[k_i \text{ is sampled}] = 1 - (1-p^K)^L - Lp^K(1-p^K)^{L-1} \quad \text{where} \quad p = 1 - \frac{1}{\pi}\arccos\frac{qk_i^T}{|q|\cdot|k_i|} \qquad (10)$$

**Data pre-processing.** Before building hash tables, MAGICPIG centers the $k_i$ vectors. As shown in Figure 2c, keys are almost always concentrated on one side of the queries, except the initial token. Random projections cannot effectively distinguish keys in this case, resulting in uniform sampled probabilities. Luckily, $\mathrm{Softmax}$ is translation invariant. Centering $(\bar{k}_i = k_i - \frac{1}{n}\sum_{i=1}^{n}k_i)$ distributed the keys better and remains computationally equivalent.

Combining Equations (9) and (10), gives a closed form of the MAGICPIG attention estimation. Assuming sample set $S$ is obtained with LSH,

$$\bar{o} = \sum_{i\in S}\frac{\exp(\frac{qk_i^T}{\sqrt{d}} - \log u_i)}{\sum_{i\in S}\exp(\frac{qk_i^T}{\sqrt{d}} - \log u_i)}v_i$$

$$u_i = 1 - (1-p_i^K)^L - Lp_i^K(1-p_i^K)^{L-1}$$

$$p_i = 1 - \frac{1}{\pi}\arccos\frac{qk_i^T}{|q|\cdot|k_i|} \qquad (11)$$

---

**Algorithm 1:** MAGICPIG Decoding

**Input:** $K, V \in R^{n\times d}, q \in R^{1\times d}$, random projectors $W \in R^{d\times(K\times L)}$, hash tables $HT$, static KV cache $K_T, V_T \in R^{t\times d}$.

*Compute hash code for new query*
$q_{\text{code}} = \textbf{Encode}(q, W)$
*Query hash tables to sample $S$ in Equation (9)*
$S = \textbf{Query}(HT, q_{\text{code}}), K_S = K[S], V_S = V[S]$
*Compute inner product for $q$ and sampled $K$*
$w_S = qK_S^T, w_T = qK_T^T$
*Compute collision probability for each hash function*
$p = 1 - \frac{1}{\pi}\arccos(w/(||q||\cdot||K_S||))$
*Compute sampling probability*
$u = 1 - (1-p^K)^L - Lp^K(1-p^K)^{L-1}$
*Compute attention output estimation*
$\bar{o} = \textbf{Softmax}(\frac{[w_S, w_T]}{\sqrt{d}} - \log([u, \mathbf{1}_t]))[V_S, V_T]$
**Return** $\bar{o}$

---

Our algorithm applies to a single attention head, see Algorithm 1. The details of **Encode**, **Query** as well as the hash table construction are described in prior work (Sadowski, 2007; Chen et al., 2020b).

## 4.4 SYSTEM CO-DESIGN

The memory size of KV cache remains a bottleneck for long-context LLM decoding, especially when GPU VRAM is limited. DRAM on the CPU side offers sufficient memory capacity with $100-200$GB/s bandwidth, which is usually $10-20\%$ of GPU VRAM bandwidth (see Figure 6a). Ideally, this gap can be mitigated by $5-10\times$ sparsity. To make CPU DRAM an *aggregated memory* for GPU, the workload must be partitioned. In our experiments $K = 9$ or $10$ and $L$ is a few hundreds.

Our system design extends prior work (He & Zhai, 2024; Aminabadi et al., 2022) by splitting LLM decoding into four parts. (1) Parameter computations, ie. all linear projectors including MLP and

$W_Q, W_K, W_V, W_O$ in the self-attention module runs on GPU. (2) Attention computation, which involves $o = \text{Softmax}(\frac{qK^T}{\sqrt{d}})V$, runs on CPU. (3) Random projections. At generation time, for each $q$, $K \times L$ random projections are conducted to obtain the hash codes. Since all heads can share the same random projectors the memory overhead is limited (400 KB in our implementation), so this step is compute-bound. Therefore, the projection is placed on GPU. (4) Retrieval. The hash codes of $q$, need to be looked up in $L$ hash tables, which is negligible computationally. However, the pre-built hash tables for $k_i$s can occupy considerable memory, making it a better fit for CPU. With the above partition, we are able to support hash tables with $K$ and $L$ beyond the scale of prior work (Kitaev et al., 2020; Chen et al., 2021; Zandieh et al., 2023) without worrying about computation for hash codes as well as the storage of hash tables.

## 5 EVALUATION

In this section, we aim to demonstrate that MAGICPIG can speed up LLM decoding while preserving high accuracy. We first present MAGICPIG's accuracy in downstream tasks, followed by our end-to-end system results showing wall-clock performance.

- In Section 5.1, we demonstrate MAGICPIG preserves high accuracy (less than $2\%$ degradation) across moderate to long context tasks with computation cost $2\% \sim 5\%$ of full attention.

- In Section 5.2, we demonstrate the efficiency of MAGICPIG, which achieves up to $5\times$ throughput improvement and 54ms decoding latency on RTX 4090 for Llama-3.1-8B-Instruct with 96K context.

- In Section 5.3, we verify the effectiveness of centering, which is of vital importance for the success of sampling. Also, we demonstrate that MAGICPIG already outperforms TopK attention in the two aggregation tasks in Figure 1, indicating that sampling indeed goes beyond TopK attention.

### 5.1 MAGICPIG PRESERVES ACCURACY

We demonstrate MAGICPIG can preserve accuracy in diverse tasks with less than $5\%$ computation.

**Setup.** Our experiments are based on Llama (AI@Meta, 2024; Dubey et al., 2024; Touvron et al., 2023) models. Three types of tasks are included, which are 3 mid-context comprehensive tasks from lm-eval-harness (Gao et al., 2021) (GSM8K-CoT (Cobbe et al., 2021), MMLU-Flan-Cot-Fewshot (Hendrycks et al., 2020) and COQA (Reddy et al., 2019)), and 6 long context tasks from (Bai et al., 2023) (QASPER (Dasigi et al., 2021), LCC, Repobench-P (Liu et al., 2023), TriviaQA (Joshi et al., 2017), PRE and TREC (Li & Roth, 2002; Hovy et al., 2001)) and 13 synthetic tasks from RULER (Hsieh et al., 2024) (with 50 examples per task).

**Baselines.** Besides full attention, Quest (Tang et al., 2024) and its variants are used as baselines. In its default setting, Quest uses a "page size" of 16, i.e. 1/16 of the full attention cost. To compare the methods fairly in the low computation budget regime, we also evaluate Quest with page size 32 and 64 and make sure at least one page is selected in every test example. The initial 4 tokens and local 64 (for LongBench (Bai et al., 2023) and RULER (Hsieh et al., 2024)) or 24 (for lm-eval-harness (Gao et al., 2021)) tokens as well as layer-$\{0,16\}$ are statically preserved. We do not use the transformations in Equation (8) in our implementations, as we do not find them to contribute to accuracy improvements.

**Cost.** The cost for the attention approximation consists of two parts: $\text{Cost}_1$ is the sampling/search cost to obtain $S$ in Equation (11), $\text{Cost}_2$ is the attention computation cost, see Equation (11). We report the ratio of number of FLOPs compared of the full attention computation. For MAGICPIG, $\text{Cost}_1 \simeq 0$ and $\text{Cost}_2$ is empirically measured for different LSH hyper-parameters. For Quest with page size $K$, $\text{Cost}_1 = \frac{1}{K}$ and $\text{Cost}_2$ is controlled manually.

**Analysis.** From Tables 1 to 3, (1) MAGICPIG preserves high accuracy (degradation less than $2\%$) for all kinds of tasks, with a computation cost of $2\% \sim 5\%$. (2) Compared with Quest, which also shows reasonable performance on long context tasks, MAGICPIG also demonstrates good performance on tasks with moderate context sizes in lm-eval-harness (Gao et al., 2021), indicating a more robust performance in general serving. (3) With LSH sampling which introduces an order of magnitude lower sampling/searching cost ($\text{Cost}_1$), MAGICPIG can achieve equivalent or better accuracy with only half of the computation cost.

### 5.2 MAGICPIG SHOWS IMPRESSIVE EFFICIENCY ACROSS VARIOUS HARDWARE SETTINGS

We show MAGICPIG can bring up to $5\times$ wall clock speed up and reduce GPU memory consumption on different models and hardware settings (A100, L20, RTX4090).

**Table 1:** Comprehensive tasks on lm-eval-harness (Gao et al., 2021). MAGICPIG significantly outperforms other methods with lower computation. The config (K,L) is hyper-parameter of LSH for MAGICPIG or page size and ratio of selected pages for Quest (Tang et al., 2024). $Cost_1$, $Cost_2$ represents cost for searching/sampling and sparse attention computation respectively.

| Methods | Config | GSM | COQA | MMLU | Avg. | $Cost_1$ | $Cost_2$ | $Cost_{total}$. |
|---|---|---|---|---|---|---|---|---|
| *Llama-2-7b-chat* | Full | 22.4 | 75.8 | 49.2 | 49.1 | 0.00 | 1.00 | 1.00 |
| MAGICPIG | (10,220) | 17.3 | 76.4 | 48.6 | **47.4** | 0.00 | 0.04 | 0.04 |
| MAGICPIG | (8,90) | 18.7 | 75.0 | 47.9 | 47.2 | 0.00 | 0.08 | 0.08 |
| Quest | (16,0.05) | 13.0 | 69.4 | 41.4 | 41.3 | 0.06 | 0.05 | 0.11 |
| Quest | (32,0.1) | 15.7 | 70.2 | 44.0 | 43.3 | 0.03 | 0.10 | 0.13 |
| *Llama-3.1-8B-Instruct* | Full | 77.6 | 78.5 | 65.2 | 73.7 | 0.00 | 1.00 | 1.00 |
| MAGICPIG | (10,220) | 72.7 | 78.1 | 62.7 | **71.2** | 0.00 | 0.03 | 0.03 |
| MAGICPIG | (8,90) | 71.0 | 78.0 | 61.3 | 70.1 | 0.00 | 0.07 | 0.07 |
| Quest | (16,0.05) | 57.9 | 64.6 | 42.5 | 55.0 | 0.06 | 0.05 | 0.11 |
| Quest | (32,0.1) | 64.5 | 65.0 | 48.0 | 59.2 | 0.03 | 0.10 | 0.13 |

**Table 2:** Long context tasks on LongBench (Bai et al., 2023). MAGICPIG preserves high accuracy with low computation. Config and cost are defined as in Table 1. Code models are only evaluated by Repobench-P and LCC.

| Methods | Config | QaS | RbP | LCC | PrE | TrC | TrQ | Avg. | $Cost_1$ | $Cost_2$ | $Cost_{total}$. |
|---|---|---|---|---|---|---|---|---|---|---|---|
| *Llama-3.1-8B-Instruct* | Full | 44.9 | 52.1 | 66.8 | 100.0 | 71.3 | 91.8 | 71.2 | 0.00 | 1.00 | 1.00 |
| MAGICPIG | (10,150) | 43.2 | 50.2 | 64.4 | 100.0 | 71.3 | 92.2 | 70.3 | 0.00 | 0.02 | 0.02 |
| MAGICPIG | (8,75) | 43.5 | 50.4 | 67.0 | 100.0 | 71.7 | 91.7 | **70.7** | 0.00 | 0.05 | 0.05 |
| Quest | (16,0.05) | 45.7 | 49.7 | 64.9 | 100.0 | 71.7 | 91.5 | 70.6 | 0.06 | 0.05 | 0.11 |
| Quest | (32,0.1) | 44.4 | 50.5 | 65.1 | 100.0 | 71.3 | 91.6 | 70.5 | 0.03 | 0.10 | 0.13 |
| *Code-Llama-13b-16K* | Full | | 58.5 | 74.7 | | | | 66.6 | 0.00 | 1.00 | 1.00 |
| MAGICPIG | (10,150) | | 56.9 | 74.0 | | | | **65.5** | 0.00 | 0.03 | 0.03 |
| Quest | (16,0.05) | | 56.4 | 74.4 | | | | 65.4 | 0.06 | 0.05 | 0.11 |

**Table 3:** Synthesized tasks on RULER (Hsieh et al., 2024). MAGICPIG preserves high accuracy with low computation. Config and cost of MAGICPIG and Quest are defined as in Table 1. We evaluate Loki with rank 32 and select 3% tokens for attention computation (Singhania et al., 2024).

| Methods | Config | 16K | 32K | 64K | 96K | Avg. | $Cost_1$ | $Cost_2$ | $Cost_{total}$. |
|---|---|---|---|---|---|---|---|---|---|
| *Llama-3.1-8B-Instruct* | Full | 94.2 | 91.5 | 86.1 | 83.0 | 88.7 | 0.00 | 1.00 | 1.00 |
| MAGICPIG | (10,150) | 91.8 | 88.9 | 84.8 | 80.0 | 86.4 | 0.00 | 0.02 | 0.02 |
| MAGICPIG | (9,120) | 93.4 | 90.6 | 84.7 | 81.5 | **87.6** | 0.00 | 0.04 | 0.04 |
| MAGICPIG | (8,75) | 92.9 | 90.2 | 84.9 | 81.7 | 87.4 | 0.00 | 0.05 | 0.05 |
| Quest | (16,0.04) | 86.3 | 85.4 | 81.9 | 74.9 | 82.1 | 0.06 | 0.04 | 0.10 |
| Quest | (32,0.06) | 84.3 | 84.0 | 80.1 | 74.4 | 80.7 | 0.03 | 0.06 | 0.09 |
| Quest | (64,0.08) | 85.2 | 84.3 | 77.0 | 74.2 | 80.2 | 0.02 | 0.08 | 0.10 |
| Loki | (32,0.03) | 80.0 | 63.6 | 61.9 | 34.7 | 60.1 | 0.12 | 0.03 | 0.15 |

**Setup.** We evaluate our system performance on 3 serving settings. (1) 80GB GPU (A100) and 34B model (CodeLlama-34B) (Rozière et al., 2024) with 16K contexts; (2) 48GB GPU (L20) and 13B model (CodeLlama-13B) (Rozière et al., 2024) with 16K contexts; (3) 24GB GPU[2] (e.g. RTX 4090) and 8B model (Llama-3.1-8B) (Dubey et al., 2024) with 96K contexts.

**Baselines.** Our baselines for (1) and (2) are full attention on GPU, and for (3) is full attention on CPU with theoretical estimated bandwidth. Our system's GPU part is implemented in native Pytorch (Paszke et al., 2019) and the CPU part in FBGEMM (Khudia et al., 2021) in bfloat16 precision. Our CPU is Intel Platinum 8480+ for A100 and Intel 8563C for L20. In the last setting, the CPU bandwidth is estimated at 150GB/s, above the empirical bandwidth we measure when running a group query attention of size 4.

**Analysis.** In Figures 7a to 7c, we demonstrate (1) MAGICPIG significantly improves decoding throughput for all three scenarios (A100: $1.5\times$, L20: $5.0\times$, RTX 4090: $3.3\times$) and can achieve a latency of 54ms for single request generation with 96K context for RTX 4090. (2) With KV cache offloading, MAGICPIG can fit much larger batches than GPU full attention baselines (over $12\times$). The ablation study of decoding throughput with different LSH hyper-parameters is presented in Table 6.

---

[2]We simulate 24GB GPU by setting memory limit with L20. As the bandwidth of L20 (864GB/s) is less than RTX 4090 (1TB/s), the real speed of our system should be slightly faster than the simulation.

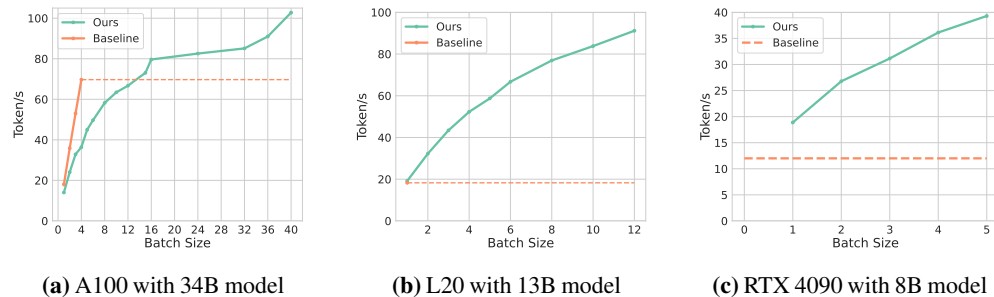

**(a)** A100 with 34B model    **(b)** L20 with 13B model    **(c)** RTX 4090 with 8B model

**Figure 7:** We evaluate MAGICPIG on three serving scenarios. **Left:** A100 serves 34B model with 16K context. MAGICPIG achieves $1.5\times$ throughput improvement. **Mid:** L20 serves 13B model with 16K context. MAGICPIG achieves $5.0\times$ throughput improvement. **Right:** Simulated RTX 4090 serves 8B model with 96K context. MAGICPIG achieves a latency of 54ms in a single request serving and can improve the throughput of baseline by up to $3.3\times$. The dashed lines denote the highest throughput of baselines. With KV cache offloading, MAGICPIG can fit a much larger batch size, which contributes to the throughput improvement.

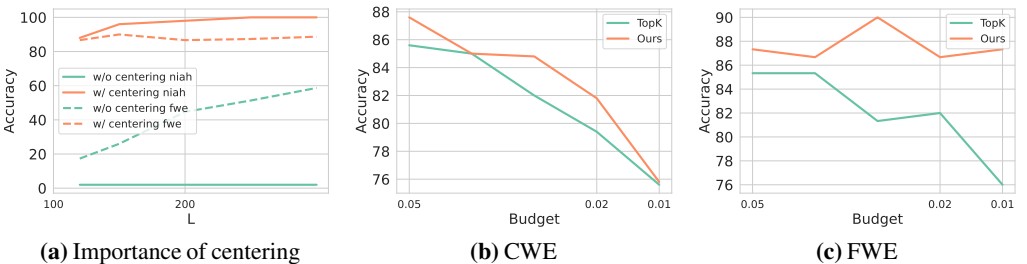

**(a)** Importance of centering    **(b)** CWE    **(c)** FWE

**Figure 8: Left:** Accuracy comparison for with and without centering. Here we fix $K$ and vary $L$ for the two settings. **Mid and Right:** In the two aggregated tasks, sampling based MAGICPIG can even beat the exact TopK attention. The experiments are done on RULER (Hsieh et al., 2024) with 16K context size.

## 5.3 ABLATION STUDY

In this section, we empirically validate our two previous observations.

**Centering is important for good performance.** In Section 4.3, we use a translation to center the keys before applying LSH sampling. Empirical results show this to be important for downstream tasks as shown in Figure 8a. Without centering, the accuracy drops to almost zero in retrieval (NIAH) and degrades to $65\%$ in FWE. We find almost none keys (less than $0.1\%$) can be sampled by query without centering, as their orientation is almost opposite as shown in Figure 2c.

**Sampling goes beyond** TopK**.** In Figures 8b and 8c, We compare the performance of MAGICPIG and TopK attention in two aggregated tasks (CWE, FWE) where TopK attention experiences significant performance degradation (Figure 1). MAGICPIG can even beat exact TopK attention in these two tasks by a margin up to $3\%$ and $8\%$ respectively, demonstrating that sampling improves the ceiling of TopK, which is impossible for a search-only algorithm.

## 6 CONCLUSION

In this work, we first present the limitation of TopK attention approximation for addressing the computational and memory challenges of long-context LLM generation. Then we show oracle sampling can go beyond TopK and introduce MAGICPIG, a novel approach that leverages LSH sampling to approximate the oracle sampling. MAGICPIG significantly reduces the workload of attention computation while preserving high accuracy across diverse tasks. MAGICPIG relies on LSH sampling and a system co-design that offloads hash tables and reduced attention computation to the CPU. Our experimental results demonstrate that MAGICPIG achieves substantial improvements in throughput and latency across multiple hardware configurations, outperforming traditional TopK attention mechanisms. The theoretical soundness, robustness, and scalability of MAGICPIG open up new opportunities in both attention approximation methods and algorithm-hardware co-design.

ACKNOWLEDGMENTS

We thank the anonymous reviewers for their helpful discussions and feedback on the paper. This work was partially supported by the National Science Foundation under grant numbers CNS-2147909, CNS-2211882, and CNS-2239351, along with gift awards from Li Auto, Amazon, Cisco, Google, Intel, Meta, Moffet AI, Oracle, Qualcomm, and Samsung.

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

## A  PROOFS FOR THEOREMS

### A.1  PROOF FOR THEOREM 3.2

*Proof.*

$$\mathbb{E}(\bar{o}) = \frac{1}{\mathcal{B}} \sum_{j=1}^{\mathcal{B}} \mathbb{E}[v_{i_j}] = \frac{1}{\mathcal{B}} \sum_{i=1}^{n} w_i v_i = o \tag{12}$$

Assume $\Sigma_1$ is the covariance matrix of $\bar{o}$, $\Sigma_2$ is the covariance matrix of $v_i$

$$\text{Tr}(\Sigma_1) = \frac{1}{\mathcal{B}} \text{Tr}(\Sigma_2) = \frac{1}{\mathcal{B}} (\mathbb{E}[||v_i||^2] - ||\mathbb{E}[v_i]||^2) = \frac{1}{\mathcal{B}} (\mathbb{E}[||v_i||^2] - ||o||^2) \tag{13}$$

$\mathbb{E}[||v_X||^2] - ||o||^2$ is a constant, so the trace of covariance matrix monotonically decreases with $\mathcal{B}$.  $\square$

### A.2  PROOF FOR THEOREM 3.3

*Proof.*

$$\mathbb{E}[|S|] = \mathbb{E}\Big[\sum_{i=1}^{n} \mathbf{1}_{i \in S}\Big] = \sum_{i=1}^{n} \mathbb{E}[\mathbf{1}_{i \in S}] = \sum_{i=1}^{n} (1 - (1-w_i)^{\mathcal{B}}) = n - \sum_{i=1}^{n} (1-w_i)^{\mathcal{B}} \tag{14}$$

Without loss of generality, let $a_i = 1 - w_i$ and $a_1 = \min_{1 \le i \le n} a_i = \epsilon$, then

$$\mathbb{E}[|S|] = n - \sum_{i=1}^{n} a_i^{\mathcal{B}} = n - a_1^{\mathcal{B}} - \sum_{i=2}^{n} a_i^{\mathcal{B}} \tag{15}$$

$$= n - \epsilon^{\mathcal{B}} - \sum_{i=2}^{n} a_i^{\mathcal{B}} \tag{16}$$

$f(x) = x^{\mathcal{B}}$ is convex function with $\mathcal{B} \ge 1$ and $x \ge 0$. Then, with Jensen's inequality, we have

$$\sum_{i=2}^{n} a_i^{\mathcal{B}} \ge (n-1)\Big(\frac{\sum_{i=2}^{n} a_i}{n-1}\Big)^{\mathcal{B}} = (n-1)\Big(\frac{(\sum_{i=1}^{n} a_i) - a_1}{n-1}\Big)^{\mathcal{B}} \tag{17}$$

$$= (n-1)(\frac{n-1-\epsilon}{n-1})^{\mathcal{B}} = (n-1)(1 - \frac{\epsilon}{n-1})^{\mathcal{B}} \tag{18}$$

Let $g(x) = (1-x)^{\mathcal{B}} + \mathcal{B}x - 1$. We can prove $g(x) \ge 0$ for any $x \in (0,1), \mathcal{B} \ge 1$. Then we have

$$\sum_{i=2}^{n} a_i^{\mathcal{B}} \ge (n-1)(1 - \frac{\epsilon \mathcal{B}}{n-1}) = n - 1 - \epsilon \mathcal{B} \tag{19}$$

Then we finally have

$$\mathbb{E}[|S|] = n - \epsilon^{\mathcal{B}} - \sum_{i=2}^{n} a_i^{\mathcal{B}} \le 1 + \epsilon \mathcal{B} \tag{20}$$

$\square$

## B  OBSERVATIONS ON ATTENTION DISTRIBUTION

To better understand the attention distribution, we study the geometry of $q, k$ and make the following three observations. (1) Key states of the initial token (also known as attention sink, denoted by $k_{sink}$) remain almost the **same** for arbitrary input. In Figure 9a, we randomly draw 32 samples from the vocabulary and measure the mutual cosine similarity of key states. Surprisingly, we find that the orientations of the key states of different input tokens are almost **identical** with a similarity $> 0.99$. (2) The orientation of the center of key states (i.e. $k_{avg} = \frac{1}{n} \sum_{i=1}^{n} k_i$) remains **stable** for different input sentences. In Figure 9b, we measure the mutual cosine similarity of $k_{avg}$ of 50 different input sentences. Although variance exists, the similarity of $k_{avg}$ is over 0.9. (3) The orientations of $k_{avg}$ and $k_{sink}$ are almost **opposite**. In Figure 9c, we find that for each head, $k_{sink}$ and $k_{avg}$ has a cosine similarity between $-0.9 \sim -0.8$.

These observations shape the geometry as shown in Figure 2c. The attention sink, which is static regardless of input, produces high sparsity in the attention distribution, whereas other parts are more uniformly distributed. Simply applying $\text{TopK}$ will place even more weight on the sink token, thus

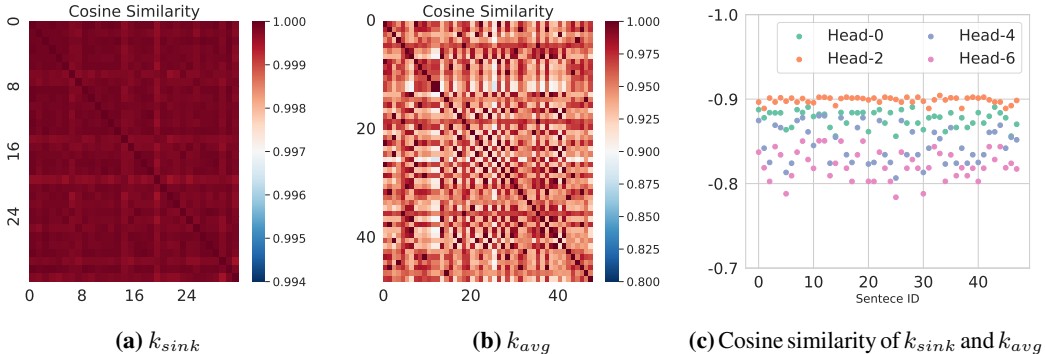

**(a)** $k_{sink}$        **(b)** $k_{avg}$        **(c)** Cosine similarity of $k_{sink}$ and $k_{avg}$

**Figure 9:** Geometric information of attention. **Left:** With arbitrary input, the orientation of $k_{sink}$ almost remains the same, with a minimum similarity $> 0.99$ across sampled inputs. **Mid:** The orientation of $k_{avg}$ is stable across various input sentences with a similarity $> 0.9$ observed. **Right:** $k_{sink}$ and $k_{avg}$ are almost opposite with similarity between $-0.9 \sim -0.8$.

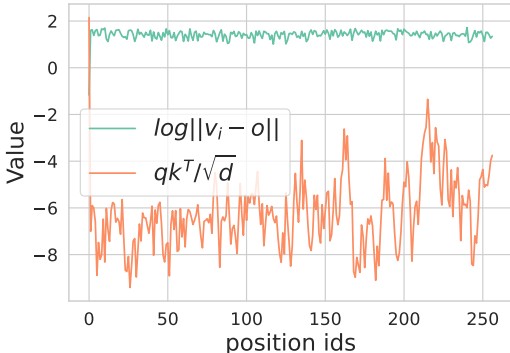

**Figure 10:** The range of fluctuation of $\log|v_i - o|$ and $\frac{qk_i^T}{\sqrt{d}}$ in a single decoding step. Compared to $\frac{qk_i^T}{\sqrt{d}}$, $\log|v_i - o|$ is stable, hence we do not consider $\log|v_i - o|$ in our proposed sampling probability.

losing contextual information. In addition, misaligning $q$ and $k$ also causes difficulty in search (Liu et al., 2024a).

## C  ORACLE SAMPLING

The optimal sampling probability to guarantee estimation is unbiased in terms of lowest variance is not directly using attention score distribution $w_i$, but $u_i' \propto w_i \|v_i\|$. However, this sampling probability is not optimal regarding downstream accuracy and efficiency. We attribute this to two reasons. First, we observe the value norm of the sink token is significantly smaller than others (Figure 11), given its lower probability of being sampled, which may influence the functionality of attention. Second, due to the same reason, $u_i' \propto w_i \|v_i\|$ is flatter than $w_i$, resulting larger computation cost (as analyzed by Theorem 3.3).

## D  SUPPLEMENTARY ANALYSIS

Figure 10 shows that compared to $\frac{qk_i^T}{\sqrt{d}}$, $\log|v_i - o|$ is stable in a decoding step.

Figure 11 shows that the norm of the value states of attention sink is smaller than others.

## E  ADDITIONAL EVALUATION

In this section, we provide additional experimental results to demonstrate that

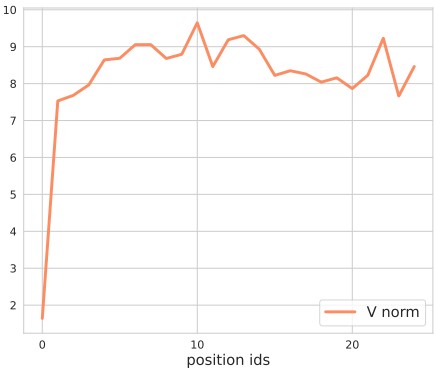

**Figure 11:** The $y$-axis is the norm of values states $\|v_i\|$ for token $i$ (on the x-axis). We observe that the value norm $\|v_0\|$ of the attention sink is significantly smaller than others.

- MAGICPIG can support longer context lengths and a wide range of LLMs (Appendix E.1).
- MAGICPIG can scale up with 70B level LLM (Appendix E.2).
- MAGICPIG improves decoding throughput with various hyper-parameters (K, L). (Appendix E.3).

## E.1 LONGER CONTEXTS

Following the setups of Table 3, we evaluate two additional models, MegaBeam-Mistral-7B-512K(`https://huggingface.co/aws-prototyping/MegaBeam-Mistral-7B-512k`) and Llama3-8B-Prolong-512K (Gao et al., 2024) with context lengths extended to 256K. The results are shown in Table 4.

**Table 4:** Synthesized tasks on RULER (Hsieh et al., 2024). MAGICPIG preserves high accuracy with extended context lengths and different models. Config and cost are defined as in Table 1.

| Methods | Config | 16K | 32K | 64K | 96K | 128K | 256K | Avg. | Cost$_1$ | Cost$_2$ | Cost$_{total}$. |
|---|---|---|---|---|---|---|---|---|---|---|---|
| *MegaBeam-Mistral-7B-512K* | Full | 91.7 | 88.1 | 83.5 | 83.7 | 83.5 | 82.5 | 85.5 | 0.00 | 1.00 | 1.00 |
| MAGICPIG | (10,150) | 89.8 | 86.5 | 81.7 | 80.7 | 81.6 | 79.0 | 83.2 | 0.00 | 0.02 | 0.02 |
| MAGICPIG | (9,120) | 90.7 | 88.5 | 82.9 | 82.4 | 82.3 | 80.1 | **84.5** | 0.00 | 0.04 | 0.04 |
| MAGICPIG | (8,75) | 90.6 | 86.4 | 82.8 | 81.6 | 82.3 | 80.8 | 84.1 | 0.00 | 0.05 | 0.05 |
| Quest | (16,0.04) | 83.3 | 83.2 | 79.3 | 78.6 | 78.5 | 78.5 | 80.2 | 0.06 | 0.04 | 0.10 |
| *Llama3-8B-Prolong-512K* | Full | 93.5 | 90.8 | 85.1 | 83.5 | 81.7 | 78.4 | 85.5 | 0.00 | 1.00 | 1.00 |
| MAGICPIG | (10,150) | 88.0 | 86.4 | 81.3 | 78.8 | 77.3 | 71.1 | 80.5 | 0.00 | 0.02 | 0.02 |
| MAGICPIG | (10,170) | 89.0 | 88.7 | 82.8 | 80.0 | 77.7 | 73.7 | 82.0 | 0.00 | 0.025 | 0.025 |
| MAGICPIG | (9,120) | 91.4 | 88.2 | 82.4 | 80.4 | 79.2 | 75.2 | **82.8** | 0.00 | 0.04 | 0.04 |
| MAGICPIG | (8,75) | 91.4 | 88.6 | 83.1 | 80.5 | 79.1 | 73.9 | 82.8 | 0.00 | 0.05 | 0.05 |
| Quest | (16,0.04) | 84.9 | 83.7 | 78.7 | 78.6 | 76.3 | 72.1 | 79.2 | 0.06 | 0.04 | 0.10 |

## E.2 SCALING UP TO LARGER MODELS

We evaluate MAGICPIG for meta-llama/Llama-3.1-70B-Instruct (Dubey et al., 2024) to demonstrate that our approach can work well with larger LLMs in Table 5.

**Table 5:** Synthesized tasks from RULER (Hsieh et al., 2024). MAGICPIG preserves high accuracy with low computation for 70B level models. 4 layers $\{0,16,32,48\}$ are preserved. Config and cost are defined as in Table 1.

| Methods | Config | 16K | 32K | 64K | Avg. | Cost$_1$ | Cost$_2$ | Cost$_{total}$. |
|---|---|---|---|---|---|---|---|---|
| *Llama-3.1-70B-Instruct* | Full | 96.4 | 94.6 | 89.2 | 93.6 | 0.00 | 1.00 | 1.00 |
| MAGICPIG | (10,135) | 95.1 | 92.3 | 86.7 | 91.7 | 0.00 | 0.016 | 0.016 |
| MAGICPIG | (10,150) | 94.8 | 93.4 | 88.2 | 92.1 | 0.00 | 0.02 | 0.02 |
| MAGICPIG | (9,110) | 95.7 | 93.5 | 88.4 | 92.5 | 0.00 | 0.034 | 0.034 |
| MAGICPIG | (9,120) | 96.0 | 94.3 | 89.1 | 93.1 | 0.00 | 0.04 | 0.04 |

## E.3 SYSTEM PERFORMANCE

In this section, we evaluate the system performance (latency, throughput) of MAGICPIG under different hyper-parameter configurations. We use Llama-3.1-8B-Instruct (Dubey et al., 2024) with 96K contexts as an example.

**Table 6:** System performance for MAGICPIG using Llama-3.1-8B-Instruct with a 96K context length under varying hyper-parameter configurations. We report the decoding latency (time between tokens, TBT) when the batch size is 1, the maximum throughput, and the throughput with a latency constraint of 200ms (Throughput$_{200ms}$ in the table). Config and cost are defined as in Table 1. The number with * means hit the memory limit of CPU.

| Config | TBT (ms) | Max Throughput (tokens/sec) | Throughput$_{200ms}$ (tokens/sec) | Cost$_{total}$. |
|---|---|---|---|---|
| (11,300) | 17.38 | 41.68* | 40.84 | 0.02 |
| (10,220) | 14.07 | 32.29* | 26.66 | 0.04 |
| (10,170) | 16.79 | 46.52* | 39.90 | 0.025 |
| (10,150) | 18.31 | 53.78 | 48.89 | 0.02 |
| (9,120) | 13.93 | 32.50 | 26.60 | 0.04 |
| (8,75) | 12.47 | 27.43 | 21.17 | 0.05 |

## F   SELECTION OF HYPER-PARAMETER (K, L)

In this section, we discuss the impact of the LSH hyper-parameter (K, L) and how to select it. First, we briefly explain what hyper-parameter (K, L) does for LSH sampling. Then, we explain the relations between (K, L) and attention computation cost and accuracy. Finally, we show how we decide the parameters by ablation studies.

### F.1   (K, L) IN LSH

In each hash table, we use K hash functions to compute the hash code of $k$ and $q$. In Simhash (Charikar, 2002), the hashing we use in MAGICPIG, the hash functions are random projections. With K random projections, we are able to partition the space (in our problem, the space is $R^{128}$) into $2^K$ subspace. If and only if $k$ and $q$ fall in the same subspace, we say they collide in this hash table. We have L hash tables in total. In MAGICPIG, if and only if $k$ and $q$ collide in at least two hash tables, $k$ is sampled by $q$. Here are some intuitions about how (K, L) will influence the LSH sampling in MAGICPIG.

- If K is too small, then we cannot partition the space well; we will sample too many $k$s, which might be far away from $q$ (in the attention problem, this means their inner production is small), increasing computation cost.

- On the other hand, if K is too large, although the quality of sampled ks will be better, the collision probability in each table will be small; thus, the number of the sampled ks will be reduced. We need to increase L to ensure that a certain number of keys are sampled and involved in the computation. However, increasing (K, L) too much will bring more memory overhead on CPU DRAM since we build L hash tables for each key-value head.

Thus, (K, L) is important because it balances computation cost, overhead, and sampling quality (which determines accuracy). Tuning (K, L) is necessary in LSH (Lv et al., 2017; Slaney et al., 2012).

### F.2   (K, L) AND MEMORY OVERHEAD

(K, L) will change two overheads brought by MAGICPIG: the memory occupied by hash tables on the CPU and extra computation for random projections (hash functions) on the GPU (as shown in Table 7).

LLM decoding is a memory-bandwidth-bound process and the majority of time is spent loading the data (parameters/KV cache) to GPU cores rather than actually doing the computation (Miao et al., 2023; Zhang et al., 2023a; Chen et al., 2024). Besides, the time-consuming part, i.e., the long-context attention computation, is moved to the CPU. Thus, the $1.8\% \sim 8.5\%$ extra computation on GPU will only make a minor difference in execution time. However, the enlarged size of hash tables prevents us from always increasing (K, L) to get more accurate results.

As shown in Table 7, under the same (K, L), the memory overhead of hash tables grows linearly with context length and the total number of key-value heads in models (which is determined by model sizes).

**Table 7:** The overhead of Locality sensitive hashing during decoding. We report the size of random projectors (on GPU) and hash tables (on CPU), the computation overhead **CO** (refers to the ratio between computation introduced by random projections in LSH and the computation of the original model's linear projections (e.g., $W_Q, W_K, W_V$, and MLP)). Notice that when the context length exceeds 64K, we need to use 32-bit integers to store the indices for the KV cache in hash tables. Llama-3.1-8B/70B-Instruct (Dubey et al., 2024) and Code-Llama-34b-16K Rozière et al. (2024) use group query attention, thus the sizes of hash tables are reduced.

| Models | (K, L) | Context length | Projectors | Hash tables | CO |
|---|---|---|---|---|---|
| *Llama-3.1-8B-Instruct* | (10, 150) | 96K | 384KB | 14GB | 3.8% |
| *Llama-3.1-8B-Instruct* | (11, 300) | 96K | 825KB | 28GB | 8.5% |
| *Llama-3.1-8B-Instruct* | (10, 150) | 64K | 384KB | 4.7GB | 3.8% |
| *Llama-3.1-70B-Instruct* | (10, 150) | 64K | 384KB | 11.8GB | 1.8% |
| *Code-Llama-13b-16K* | (10, 150) | 16K | 384KB | 7.3GB | 5.2% |
| *Code-Llama-34b-16K* | (10, 150) | 16K | 384KB | 1.8GB | 2.2% |

### F.3 (K, L) AND COMPUTATION COST/BUDGET

In summary, increasing K will make the budget[3] smaller, and increasing L will increase the budget.

Theoretically, as introduced in Section 4.3, in our approach, the key $k_i$ is sampled only if at least two hash tables exist where $k_i$ shares the hash value with query q. With the assumption that $k_i$ is well-distributed (In each hash table out of L, each hash value corresponds to roughly the same number of $k_i$s), the ratio of retrieved $k_i$s can be estimated with

$$\mathcal{B}/n = 1 - (1 - 0.5^K)^L - L \times 0.5^K (1 - 0.5^K)^{L-1} \tag{21}$$

where $n$ is the context length. Here, we estimate the collision probability of $k_i$ and $q$ in a single hash table as $0.5^K$.

Empirically, the ratio of retrieved keys and values ($\mathcal{B}/n$) might differ from the above estimation since the data is not perfectly distributed. We present the empirically measured budget in Table 8.

**Table 8:** Empirical measured budget/cost for different (K, L).

| K / L | 75 | 100 | 120 | 150 | 200 | 300 |
|---|---|---|---|---|---|---|
| 7 | 14% | 21% | 27% | 35% | 48% | 66% |
| 8 | 5% | 8% | 11% | 15% | 22% | 36% |
| 9 | 1.6% | 2.7% | 4% | 5.4% | 8.5% | 15.4% |
| 10 | 0.5% | 0.9% | 1.5% | 2% | 3% | 6% |
| 11 | 0.15% | 0.3% | 0.5% | 0.6% | 1% | 2% |

### F.4 (K, L) AND ACCURACY

There are no naive relations between (K, L) and downstream accuracies since (K, L) not only influences sampling quality but also the computation budget. One safe way to discuss the relation between (K, L) and accuracy is: Fixing the computation budget, larger (K, L) will potentially produce higher accuracy, since the sampling quality is higher. Our experimental results show that,

- Increasing (K, L) can significantly improve accuracy in relatively longer contexts Table 9.

**Table 9:** We show the effectiveness of larger hash tables for longer contexts by evaluating MegaBeam-Mistral-7B-512K on RULER (Hsieh et al., 2024). With the same computation cost ($\sim 2\%$), config (11, 300) achieves higher accuracy compared to (10, 150).

| (K, L) | 16K | 128K | 256K |
|---|---|---|---|
| Full | 91.7 | 83.7 | 82.5 |
| (10, 150) | 89.8 | 80.7 | 79.0 |
| (11, 300) | 90.6 | 83.3 | 81.9 |

- Same set of (K, L) can generalize to larger LLMs Table 10.

---

[3]Cost$_2$ in Tables 1 to 3

**Table 10:** 8B and 70B models on RULER (Hsieh et al., 2024) 64K.

| Models/Config | Full | (10, 150) | (10, 135) | (9, 120) | (9, 110) |
|---|---|---|---|---|---|
| Llama-3.1-8B-Instruct | 86.1 | 84.8 | 83.6 | 84.7 | 84.7 |
| Llama-3.1-70B-Instruct | 89.2 | 88.2 | 86.7 | 89.1 | 88.4 |

### F.5 HOW TO SELECT (K, L)

Finding the optimal (K, L) for high accuracy as well as efficiency is a long-standing problem in LSH. Similar to the traditional hyper-parameter tuning process in machine learning, K, and L are configured offline based on data subsets. In LSH, K is a more sensitive hyper-parameter than L. A slight change of K can drastically influence the number of retrieved items (i.e., budget/cost) and quality. In MAGICPIG, K=8-10 is manually determined by ablations on small-scale tasks and found to be effective across various models and tasks; then, we adjust L to obtain the wanted computation cost/budget.

Here, we present two ablations to demonstrate the selection of K in Tables 11 and 12.

**Table 11:** Fixing the budget/cost to $4\%$, we ablation the performance of different (K, L) on RULER (Hsieh et al., 2024) 16K.

| Models/Config | Full | (10, 240) | (9, 120) | (8, 65) | (7, 35) |
|---|---|---|---|---|---|
| Llama-3.1-8B-Instruct | 94.2 | 94.2 | 92.8 | 92.3 | 88.5 |

**Table 12:** Fixing L as 120, we ablation the performance of different K on RULER (Hsieh et al., 2024) 16K for Llama-3.1-8B-Instruct.

| (K, L) | Full | (10, 120) | (9, 120) | (8, 120) | (7, 120) |
|---|---|---|---|---|---|
| Cost | 1.0 | 0.012 | 0.04 | 0.11 | 0.27 |
| Accuracy | 94.2 | 92.8 | 92.8 | 94.1 | 94.3 |

If we want the computation cost to be below $5\%$ and L below 200 (to reduce memory overhead in the CPU), then K=8-10 is a reasonable choice. Unlike K, L is not that sensitive. We select L based on the following principle after determining K: we can allow the computation cost to be smaller for larger K since the sampling is more precise. This is why we choose to use (8, 75), (9, 120), and (10, 150).

It's worth pointing out that tuning (K, L) is a challenging and long-standing problem in LSH, and we only give an example of practice in MAGICPIG. More advanced hashing algorithms (such as Cross-polytope (Andoni et al., 2015) or data-dependent ones (Andoni & Razenshteyn, 2015)) can improve the trade-off between memory overhead and accuracy. We leave it as a future direction.

## G TOPK VS. SAMPLING

In this section, we provide an intuitive understanding of how sampling can work better than TopK. TopK only captures the ranking information when estimating attention output. In contrast, sampling considers the entire data distribution (i.e., the attention score after Softmax).

Here is an example. Imagine a zoo with 100 animals: 10 elephants, 10 pigs, 10 tigers, and 70 other unique animals. The daily food consumption for each group is as follows:

- **Elephants**: 50 lb/day each
- **Pigs**: 20 lb/day each
- **Tigers**: 10 lb/day each
- **Other unique animals**: 1 lb/day each

To compute the true average daily food consumption per animal in the zoo:
$$\text{True Average} = \frac{(10 \times 50) + (10 \times 20) + (10 \times 10) + (70 \times 1)}{100} = 8.7 \text{lb}.$$

If we use a **Top-K approach** (e.g., selecting the top 10 animals based on the numbers of animals), we include elephants, pigs, tigers, and 7 randomly selected animals from the unique ones. The estimated

average is:

$$\text{TopK Average} = \frac{(10 \times 50) + (10 \times 20) + (10 \times 10) + (7 \times 1)}{37} = 22\text{lb}.$$

This overestimates the average because it disproportionately weights high-consumption animals.

Instead, we perform **sampling with replacement** from the animal distribution, proportional to their numbers. The probabilities for each group are:

$$\text{Sampling Probabilities} = [0.1, 0.1, 0.1, 0.01 \times 70],$$

where 0.1 represents the probabilities for elephants, pigs, and tigers (10/100 each), and 0.01 corresponds to each unique animal (1/100).

Perform 10 random draws. A possible sampling outcome could be: [elephant, pig, tiger, other, other, other, other, other, other, other]. The corresponding daily food estimate is:

$$\text{Sample Estimate} = \frac{50 + 20 + 10 + (7 \times 1)}{10} = 8.7\text{lb}.$$

This estimate is unbiased, meaning the expected value of the estimates equals the true average (8.7 lb). While there is variance across individual trials, the **standard deviation (std)** can be calculated as 4.7 lb for a 10-sample budget.

Increasing the sampling budget reduces variance. For example, with 20 samples, the **std** decreases to 3.4 lb. Meanwhile, Top-K with a budget of 20 adds 17 unique animals, yielding:

$$\text{TopK Average (K=20)} = \frac{(10 \times 50) + (10 \times 20) + (10 \times 10) + (17 \times 1)}{47} = 17\text{lb}.$$

Again, the Top-K estimate remains biased, significantly overestimating the average.

Note that this is intended as an intuitive example. For a detailed and formal derivation of the sampling methodology, please refer to Kloek & Van Dijk (1978); Owen (2013); Lohr (2021).

## H  LIMITATIONS AND FUTURE WORK

MAGICPIG stores the offloaded KV cache and hash tables on CPU DRAM, which is unsuitable for serving scenarios with insufficient DRAM. KV cache quantization methods like QServe (Lin et al., 2024) and KIVI (Liu et al., 2024b) can help to reduce the KV cache memory. Currently, another limitation is that, we have not implemented MAGICPIG in prefilling stage, which is also an important direction in long context LLM serving. Applying more advanced LSH algorithms, such as Cross-polytope hash (Andoni et al., 2015), can reduce the size of hash tables while improving estimation accuracy. Building CPU-GPU pipelines (He & Zhai, 2024) and leveraging the new avx512_bf16 features of CPUs will improve efficiency. For higher-end GPUs with sufficient HBM, leveraging LSH to accelerate GPU attention computation is also an interesting topic, as GPU-friendly LSH algorithms and efficient GPU kernels (Nguyen Mau & Inoguchi, 2020; Pan et al., 2022) are required to do sampling. Besides, how to automatically tune the LSH hyper-parameter (K, L) (Lv et al., 2017) is also an interesting future work.

