# OpenReview forum: "MagicPIG: LSH Sampling for Efficient LLM Generation"
_ICLR.cc/2025/Conference — ICLR 2025 Spotlight_

### Official Review · Reviewer_qU5o · 2024-10-31

**Soundness:** 3
**Presentation:** 4
**Contribution:** 4
**Rating:** 8
**Confidence:** 3

**Summary:**

This paper introduced a novel method dubbed "MagicPIG" to reduce the computation cost of self-attention in long context. Specifically, MagicPIG utilizes Locality-sensitive hashing to approximate the attention score distribution and estimate the attention output. While not decreasing the overall cache size required to store Keys and Values, MagicPIG sampled only a fraction of Keys and Values to calculate the attention scores, reducing the overall computation cost.

**Strengths:**

This paper is exceptionally well-written and clearly presented, making it tremendously helpful for understanding complex topics. Concepts, definitions, and proofs are structured logically, with clear and concise writing. The proposed approach is intuitive and relatively straightforward, with much of the intuition supported by prior explanations. Additionally, the empirical results are strong.

**Weaknesses:**

I have a few questions:
1. what is the intuition for selecting (K, L) for the hash table size?
2. For Table 1/2/3, why is latency not included in the comparison?
3. Does the author believe further improvement can be made by combining this approach with PEFT?

**Questions:**

See weaknesses

---

> ### Author Response · Authors · 2024-11-22
>
> Thank you very much for your insightful review and constructive suggestions. We are glad that the reviewer found our work **well-written, intuitive, and empirically strong**. We have tried to address your questions carefully. We hope the reviewer will consider raising your score in light of our response.
>
> ## Q1: selection of K, L
>
> Thank you for raising this question. $\text{\textcolor{blue}{Selecting optimal (K, L) is a challenging and long-standing problem in LSH}}$. **First**, we briefly explain what hyper-parameter (K, L) means for LSH sampling (just for reference). **Second**, we explain the relations between (K, L) and attention computation cost and accuracy. **Finally**, we show how we decide the parameters by ablation studies.   A more detailed discussion is added in  $\text{\textcolor{blue}{Appendix E, Pg. 18-20}}$ and also presented in "reply to all reviewers".
>
> ### **What (K, L) do in LSH.**
> In each hash table, we use K hash functions to compute the hash code of $k$ and $q$.  In Simhash, i.e., the hashing we use in MagicPIG, the hash functions are random projections. With K random projections, we are able to partition the space (in our problem, the space is $R^d$) into $2^K$ subspace. If and only if $k$ and $q$ fall in the same subspace, we say $k$ and $q$ collide in this hash table.  We have L hash tables in total. In MagicPIG, if and only if $k$ and $q$ collide in at least two hash tables, $k$ is sampled/retrieved by $q$.  Intuitively,
> - **if K is too small**, we cannot partition the space well. We will sample too many $k$s, which might be far away from q (in the attention problem, this means their inner production is small), resulting in an increase in computation cost.
> - On the other hand, **if K is too large**,  although the quality of sampled $k$s will be better, the collision probability in each table will be small; thus, the number of sampled $k$s will be reduced. We need to increase L to ensure that at least a certain amount of keys are sampled and involved in the computation.  However, increasing (K, L) too much will bring more memory overhead on CPU DRAM, since we build L hash tables for each key-value head.
>
> ### **(K, L) and computation cost/budget.**
> In summary, increasing K will make the budget smaller, and increasing L will increase the budget.
> - $\text{\textcolor{blue}{(Theoretically)}}$ As introduced in Section 4.3, in our approach, the key $k_i$ is sampled only if at least two hash tables exist where $k_i$ shares the hash value with query $q$. With the assumption that $k_i$ is well-distributed (In each hash table out of L, each hash value corresponds to roughly the same number of $k_i$s), the ratio of retrieved $k_i$s can be estimated with
> $ \mathcal{B} / n = 1 - (1 - 0.5^K)^L - L 0.5^K (1 - 0.5^K)^{(L-1)} $,
> where $n$ is the context length, here, we estimate the collision probability of $k_i$ and $q$ in a single hash table as $0.5^K$.
>
> - $\text{\textcolor{blue}{(Empirically)}}$ The ratio of retrieved keys and values ($\mathcal{B} / n$) might differ from the above estimation since the data is not perfectly distributed.  In our experiments, after fixing (K, L), we empirically measure the number of keys and values accessed each time and report their averages. We present the empirically measured budget below,
>
>
> | K / L | 75 | 100 | 120 | 150 | 200 | 300|
> | ---|---|---|---|---|---|---|
> |  7      | 14% | 21%| 27% |35%| 48% | 66%|
> |  8      |  5%|  8% | 11%| 15% | 22% | 36%|
> |  9      |  1.6% | 2.7%| 4% | 5.4%| 8.5% | 15.44%|
>  |  10    | 0.5% |  0.9% | 1.2% | 2% | 3%| 6%|
> |   11   | 0.15% | 0.3%| 0.5%|  0.6% | 1%| 2%|
>
>
> ### **(K, L) and accuracy.**
>
> There is no simple relation between (K, L) and downstream accuracy since (K, L) not only influences sampling quality but also influences the computation budget. One safe way to discuss the relation between (K, L) and accuracy is: **Fixing the computation budget, larger (K, L) will potentially produce higher accuracy since the sampling quality is higher.**
>
> Our experimental results show that,
> - $\text{\textcolor{blue}{Increasing (K, L) can significantly improve accuracy in relatively longer contexts}}$
> Model: MegaBeam-7B-512K
> |   Methods  |  Config  |  16K |128K | 256K | Total Cost |
> |    ------        | -----       | -----|-----  |  -----  |  -----  |
> | |  Full      |  91.7 |83.7   | 82.5|   1.0 |
> | MagicPIG  |(10,150) |  89.8 | 80.7|  79.0|   0.02|
> | MagicPIG  |(11,300)  |  90.6    |83.3|  81.9|   0.02|
>
> - $\text{\textcolor{blue}{Same set of (K, L) can generalize to larger LLMs}}$
>
> |   Models / Config  |  Full | (10,135) |(10, 150) | (9, 110) | (9, 120) |
> |    ------        | -----       | -----|-----  |  -----  |  -----  |
> |    Llama3.1-8B-Instruct   |  86.1      |  83.6 |84.8   | 84.7|   84.7 |
> |    Llama3.1-70B-Instruct   |  89.1      |  86.7 |88.2   | 88.4|  89.1 |

---

> ### Author Response · Authors · 2024-11-22
> **Q1 (Part2)**
>
> ### **How to select (K, L).**
> **Finding the optimal (K, L) for high accuracy and efficiency is a long-standing problem in LSH**.  Like the traditional hyperparameter tuning process in machine learning, K and L are configured offline based on data subsets. In LSH, **K is a more sensitive hyperparameter than L**.  A slight change of K can drastically influence the number of retrieved items (i.e., budget) and quality. In MagicPIG, K=8-10 is **manually** determined by ablations on small-scale tasks and found to be effective across various models and tasks. Then, we adjust L to obtain the desired computation cost/budget.
>
> Here,  we present two ablations to demonstrate the selection of K.
>
> Model: Llama-3.1-8K-Instruct; Task: RULER + 16k; Full model accuracy: **94.2**
>
> - $\text{\textcolor{blue}{Exp1: Vary L and fix the computation cost/budget}}$
>
> | K       |  L |  Accuracy |  cost |
> |   ----- | -----| -----     | -----   |
> |  10    |  240|  94.2   | 4%|
> |   9     |  120| 92.8 | 4%|
> |   8     |  65  | 92.3 | 4%|
> |   7     |  35  |   88.5  |  4%|
>
>
> - $\text{\textcolor{blue}{Exp2: Fix L as 120 and vary K (the cost/budget will also vary)}}$
>
> | K       |  L |  ACC |  cost |
> |   ----- | -----| -----     | -----   |
> |  11   |   120| 60.2  |    0.5%|
> |  10    |  120| 87.3 | 1.2%|
> |   9     |  120|92.8 | 4%|
> |   8     |  120| 94.1 | 11%|
> |   7     |  120 | 94.3 | 27%|
>
> If we want the computation cost to be below 5% and L below 200 (to reduce memory overhead in the CPU), then K=8-10 is a reasonable choice. Unlike K, L is not that sensitive. We select L **based on the following principle** after determining K: for larger K, we can allow the computation cost to be smaller. This is why we choose to use (8, 75), (9, 120), and (10, 150).
>
> It’s worth pointing out that tuning (K, L) is a challenging problem in LSH [1], and we only give a simple example in MagicPIG. More advanced hashing algorithms (such as Cross-polytope [2] or data-dependent ones [3]) can improve the trade-off between memory overhead and accuracy. We leave it as a future direction.
>
> [1] Qin Lv, William Josephson, Zhe Wang, Moses Charikar, and Kai Li. 2017. Intelligent probing for locality sensitive hashing: multi-probe LSH and beyond. Proc. VLDB Endow. 10, 12 (August 2017), 2021–2024. https://doi.org/10.14778/3137765.3137836
>
> [2] Kitaev, Nikita, Łukasz Kaiser, and Anselm Levskaya. "Reformer: The efficient transformer." arXiv preprint arXiv:2001.04451 (2020).
>
> [3] Andoni, Alexandr, and Ilya Razenshteyn. "Optimal data-dependent hashing for approximate near neighbors." Proceedings of the forty-seventh annual ACM symposium on Theory of computing. 2015.

---

> ### Author Response · Authors · 2024-11-22
>
> ## Q2: Why are latencies not reported in Table 1/2/3?
>
> In Section 5.1, the cost metric is a fraction of the FLOPs performed with full attention. This is a more objective measure of complexity because it is **independent of hardware or implementation**. Our baselines, e.g., Quest and Loki (newly added), **do not have CPU implementations**, and a naive implementation can be inefficient. In Section 5.2, we provide the runtime equivalents of a few settings. For example, LongBench roughly has an average context size of 16K, and RULER ranges from 16K to 96K in our experiments.
>
> ## Q3: Combination with parameter-efficient fine-tuning?
> Currently, MagicPIG is optimized for inference, i.e., forward passes with 1-token batches, but we believe it is a very promising future direction to be combined with PEFT:
>
> - It could be extended to accelerate PEFT. Since building the hash tables is very efficient, building them for each training sequence should be possible. In fact, LSH has been studied to reduce computation and accelerate the training of linear layers [1].
> - Leveraging PEFT to make attention estimation more accurate (e.g., Locret [2], which trains LLMs to discard the KV cache, might be related to this problem) is also promising.
>
> [1] Chen, Beidi, et al. "Slide: In defense of smart algorithms over hardware acceleration for large-scale deep learning systems." arXiv preprint arXiv:1903.03129 (2019).
>
> [2] Huang, Yuxiang, et al. "Locret: Enhancing Eviction in Long-Context LLM Inference with Trained Retaining Heads." arXiv preprint arXiv:2410.01805 (2024).

---

> > ### Comment · Reviewer_qU5o · 2024-12-02
> > **response to authors**
> >
> > I thank the authors for their response. I shall maintain my score.

---

### Official Review · Reviewer_Q8fb · 2024-11-03

**Soundness:** 3
**Presentation:** 3
**Contribution:** 3
**Rating:** 6
**Confidence:** 4

**Summary:**

The authors introduce MAGICPIG, a heterogeneous system leveraging LSH (Locality-Sensitive Hashing) sampling to estimate a complete attention distribution, overcoming limitations of traditional Top-K sparse attention methods, which can underperform in certain downstream tasks.

**Strengths:**

1.	MAGICPIG addresses shortcomings of traditional Top-K attention methods in LLMs, which often assume sparsity and suffer in some downstream applications. Using LSH-based sampling, MAGICPIG more accurately estimates the attention distribution, mitigating the bias found in Top-K approximations. The approach is backed by theoretical guarantees and empirical evidence, underscoring its effectiveness in sparse attention acceleration.

2.	MAGICPIG overcomes GPU VRAM constraints by offloading parts of the computation, including hash table operations, to the CPU. This approach is pivotal for scaling LLMs with LSH-based sampling in resource-constrained, practical environments.

**Weaknesses:**

1.	While the authors discuss CPU-GPU collaboration, they provide limited data on the effects of PCIe bandwidth and CPU-GPU data transfer overhead. This omission may hinder understanding MAGICPIG’s real-world performance across different hardware configurations.

2.	The paper lacks a detailed analysis of the overhead associated with hash tables. As noted by the authors, hash tables could introduce significant memory and computational costs. Therefore, a more thorough evaluation of these overheads would better illustrate the trade-offs of the proposed method.

**Questions:**

1.	Is the size of the hash table related to model size and sequence length? How does the size of the hash table affect the performance?

2.	What is the time overhead of constructing hash tables, and which factors influence this overhead?

---

> ### Author Response · Authors · 2024-11-22
>
> Thank you very much for your insightful review and constructive suggestions. We are glad the reviewer found our work **pivotal** and **empirically effective**. We have tried to address your questions carefully. We hope the reviewer will consider raising your score in light of our response.
>
> ## W1: While the authors discuss CPU-GPU collaboration, they provide limited data on the effects of PCIe bandwidth and CPU-GPU data transfer overhead. This omission may hinder understanding MAGICPIG’s real-world performance across different hardware configurations.
>
> Thank you for your question.
> In our evaluation, the communication time is between 6-10 ms,  while the latency for each decoding iteration is between 90-400ms, depending on model architectures, batch sizes, and sequence lengths. Therefore, the overhead of CPU-GPU data transfer is not an important bottleneck most of the time.
>
> Here, we present two example breakdowns of our system's execution time.
>
> $\text{\textcolor{blue}{Model: Llama-3.1-8B-Instruct; Context Length: 96K}}$
> |   Batch size  |  CPU  |  Data Transfer | GPU | Total time|
> |    ------        | -----       |   -----  |  -----  |  -----  |
> | 1  |  64ms  | 6ms |  40ms | 110ms |
> | 4  |  128ms| 6ms |    43ms|  177ms|
>
> $\text{\textcolor{blue}{Model: CodeLlama-34B; Context Length: 16K }}$
> |  Batch size  |  CPU  |  Data Transfer | GPU |Total time|
> |    ------        | -----       |   -----  |  -----  |  -----  |
> |       2           |       31ms      |  8ms |   56ms  | 95ms|
> |      18         |        190ms    |    10ms      |   66ms | 266ms|
>
> Since we only transfer the query, query’s hash code, and attention output through PCIE, which is a very small tensor (e.g., less than 1MB per layer) compared to the KV cache, **the communication time is mainly determined by the copy launching latency, not bottlenecked by the PCIE bandwidth, so it is almost not influenced by batch size (but is influenced by how many times we call the device to host memory copy functions)**.  This feature makes MagicPIG work better with large batch sizes than small batch size. Using **page-locked-memory** can potentially reduce copy launching latency, thus improving MagicPIG’s performance in small batch size. We leave this optimization in the future plan.
>
> Some explanations: For the CPU execution part, we have not implemented the thread scheduler. We use one open-mp thread to compute one attention head. When the batch size is small, some CPU cores might be idle, resulting in a relatively long execution time compared to a large batch size. Our preliminary result shows that, by simply splitting the workload in one attention head to multiple cores, we can reduce the latency $\text{\textcolor{blue}{from 110 ms to around 75 ms }}$ for Llama-3.1-8B + 96K context.
>
> ## W2:The paper lacks a detailed analysis of the overhead associated with hash tables. As noted by the authors, hash tables could introduce significant memory and computational costs. Therefore, a more thorough evaluation of these overheads would better illustrate the trade-offs of the proposed method.
>
> Thanks for the suggestions. We add a detailed discussion of memory/computation overhead of hash tables in  $\text{\textcolor{blue}{Appendix E.2, Pg.19, Table 6 }}$.
>
> |Models  | (K, L) | Context length | Size of Projectors | Size of Hash tables |  GPU Extra Computation|
> | --- | --- | --- | --- | --- | --- |
> |Llama-3.1-8B-Instruct | (10, 150) | 96K | 384KB | 14GB| 3.7%|
> |Llama-3.1-8B-Instruct | (11, 300) | 96K | 825KB | 28GB| 8.5%|
> |Llama-3-8B-512K | (10, 150) | 256K | 384KB | 37GB| 3.7%|
> |Llama-3-8B-512K | (11, 300) | 256K | 825KB | 74GB| 8.5%|
> |Llama-3.1-70B-Instruct | (10, 150) | 96K | 384KB | 70GB| 1.8%|
>
> As LLM decoding is a **memory-bandwidth-bound process**, the major time is spent on loading the data (parameters/KV cache) to GPU cores rather than actually doing the computation [1][2][3][4]. Besides, the time-consuming part, i.e., the long-context attention computation, is moved to the CPU in our system. Thus, the 1.8%-8.5% extra computation on GPU will only make a minor difference in E2E execution time. However, the enlarged **size of hash tables** prevents us from always increasing (K, L) to get more accurate results.
>
> [1] Miao, Xupeng, et al. "SpecInfer: Accelerating Generative Large Language Model Serving with Tree-based Speculative Inference and Verification." arXiv preprint arXiv:2305.09781 (2023).
>
> [2] Chen, Zhuoming, et al. "Sequoia: Scalable, robust, and hardware-aware speculative decoding." arXiv preprint arXiv:2402.12374 (2024).
>
> [3] Liu, Xiaoxuan, et al. "Online speculative decoding." arXiv preprint arXiv:2310.07177 (2023).
>
> [4] Yuan, Zhihang, et al. "Llm inference unveiled: Survey and roofline model insights." arXiv preprint arXiv:2402.16363 (2024).

---

> ### Author Response · Authors · 2024-11-22
>
> ## Q1(1): Is the size of the hash table related to model size and sequence length?
>
> Yes, the size of the hash table is related to model size and sequence length. Under the same LSH hyper-parameter (K, L), the memory overhead of hash tables grows linearly with context length and the total number of key-value heads in models (which is determined by model sizes).
>
> - **Model size.**  **First**, larger models will usually have more layers and more key value heads. Since we build L hash tables for each individual key-value head, if we use the same LSH hyper-parameter (K, L), the total memory occupied by hash tables will be larger for larger models.  **Second**, our empirical results show that the same set of LSH hyper-parameters (K, L), can generalize to larger models (added in $\text{\textcolor{blue}{Appendix D.2, Pg.18, Table 6 and Appendix E.4, Pg. 20, Table 9}}$).
>
> |   Models / Config  |  Full | (10,135) |(10, 150) | (9, 110) | (9, 120) |
> |    ------        | -----       | -----|-----  |  -----  |  -----  |
> |    Llama3.1-8B-Instruct   |  86.1      |  83.6 |84.8   | 84.7|   84.7 |
> |    Llama3.1-70B-Instruct   |  89.1      |  86.7 |88.2   | 88.4|  89.1 |
>
> - **Sequence length.** **First**, longer sequence length will make the hash table larger, since we need to store more indices. **Second**, increasing hyper-parameter (K, L) of LSH (which will enlarge the size of hash tables) can lead to better performance in longer context situations. We have added empirical evidence in $\text{\textcolor{blue}{Appendix E.4, Pg.20, Table 8}}$.
>
> Model: MegaBeam-7B-512K
> |   Methods  |  Config  |  16K |128K | 256K | Total Cost |
> |    ------        | -----       | -----|-----  |  -----  |  -----  |
> | |  Full      |  91.7 |83.7   | 82.5|   1.0 |
> | MagicPIG  |(10,150) |  89.8 | 80.7|  79.0|   0.02|
> | MagicPIG  |(11,300)  |  90.6    |83.3|  81.9|   0.02|
>
> We present details of memory/computation overhead of hash tables in $\text{\textcolor{blue}{Appendix E.2, Pg.19, Table 6}}$.
>
> ## Q1(2): How does the size of the hash table affect the performance?
>
> The relation between (K, L) (the LSH hyper-parameter, which decides the sizes of hash tables under fixed context length and model architecture) and accuracy is complicated as we analyze in **Notes[1,2,3]** below. One safe way to say is:  **Fixing the computation budget, larger (K, L) will potentially produce higher accuracy since the sampling quality is higher**.  In fact, there is **no simple relationship** between (K, L) and downstream accuracy since (K, L) not only influences **sampling quality** but also influences the **computation budget**.
>
> We provide some notes on how (K, L) influences the LSH process and the attention computation cost/budget. A more detailed discussion is added in $\text{\textcolor{blue}{Appendix E (Pg. 18-20)}}$ and also presented in  "reply to all reviewers".
>
> **Note1: What (K, L) do with LSH.**  In each hash table, we use K hash functions to compute the hash code of $k$ and $q$.  In Simhash, i.e., the hashing we use in MagicPIG, the hash functions are random projections. With K random projections, we are able to partition the space (in our problem, the space is $R^d$) into $2^K$ subspace. If and only if $k$ and $q$ fall in the same subspace, we say $k$ and $q$ collide in this hash table.  We have L hash tables in total. In MagicPIG, if and only if $k$ and $q$ collide in at least two hash tables, $k$ is sampled/retrieved by $q$.  Intuitively,
> - if K is too small, then we cannot partition the space well, we will sample too many $k$s, which might be actually far away from $q$ (in the attention problem, this means their inner production is small), resulting in an increase in computation cost.
> - On the other hand, if K is too large,  although the quality of sampled $k$s will be better, the collision probability in each table will be small thus the number of the sampled $k$s will be reduced. We need to increase L to make sure that at least a certain amount of keys are sampled and involved in the computation.  However, increasing (K, L) too much will bring more memory overhead on CPU DRAM, since we build L hash tables for each key-value head.

---

> ### Author Response · Authors · 2024-11-22
> **Q1(2) (Part2)**
>
> **Note2: (K, L) and computation cost/budget.** In summary, increasing K will make the budget smaller, and increasing L will increase the budget.
>  - $\text{\textcolor{blue}{(Theoretically)}}$  As introduced in $\text{\textcolor{blue}{Section 4.3 (Pg. 7)}}$, in our approach, the key $k_i$ is sampled only if at least two hash tables exist where $k_i$ shares the hash value with query $q$. With the assumption that $k_i$ is well-distributed (In each hash table out of L, each hash value corresponds to roughly the same number of $k_i$s), the ratio of retrieved $k_i$s can be estimated with
> $\mathcal{B} / n = 1 - (1 - 0.5^K)^L - L \times 0.5^K (1 - 0.5^K)^{(L-1)} $, where $n$ is the context length, here, we estimate the collision probability of $k_i$ and $q$ in a single hash table as $0.5^K$.
>
> - $\text{\textcolor{blue}{(Empirically)}}$ The ratio of retrieved keys and values $\mathcal{B} / n$ might differ from the above estimation since the data is not perfectly distributed.  In our experiments, after fixing (K, L), we **empirically** measure the number of keys and values accessed each time and report their averages. We present the empirically measured budget below,
>
> | K / L | 75 | 100 | 120 | 150 | 200 | 300|
>  | ---|---|---|---|---|---|---|
> |  7      | 14% | 21%| 27% |35%| 48% | 66%|
> |  8      |  5%|  8% | 11%| 15% | 22% | 36%|
> |  9      |  1.6% | 2.7%| 4% | 5.4%| 8.5% | 15.44%|
>   |  10    | 0.5% |  0.9% | 1.2% | 2% | 3%| 6%|
>  |   11   | 0.15% | 0.3%| 0.5%|  0.6% | 1%| 2%|
>
> **Note3: (K, L) and accuracy.**  There is no simple relationship between (K, L) and downstream accuracy since (K, L) not only influences sampling quality but also influences the computation budget.  Fixing the computation budget, larger (K, L) will potentially produce higher accuracy, since the sampling quality is higher. In addition, our experimental results show that,
>
> - $\text{\textcolor{blue}{Increasing (K, L) can significantly improve accuracy in relatively longer contexts}}$
>
> Model: MegaBeam-7B-512K
> |   Methods  |  Config  |  16K |128K | 256K | Cost |
> |    ------        | -----       | -----|-----  |  -----  |  -----  |
> | |  Full      |  91.7 |83.7   | 82.5|   1.0 |
> | MagicPIG  |(10,150) |  89.8 | 80.7|  79.0|   0.02|
> | MagicPIG  |(11,300)  |  90.6    |83.3|  81.9|   0.02|
>
> - $\text{\textcolor{blue}{Same set of (K, L) can generalize to larger LLMs}}$
>
> |   Models / Config  |  Full | (10,135) |(10, 150) | (9, 110) | (9, 120) |
> |    ------        | -----       | -----|-----  |  -----  |  -----  |
> |    Llama3.1-8B-Instruct   |  86.1      |  83.6 |84.8   | 84.7|   84.7 |
> |    Llama3.1-70B-Instruct   |  89.1      |  86.7 |88.2   | 88.4|  89.1 |

---

> ### Author Response · Authors · 2024-11-22
>
> ## Q2: What is the time overhead of constructing hash tables, and which factors influence this overhead?
>
> Thanks to the low construction time of the hash table and our strategy of **overlapping the hash table construction time with the prefilling time,** the e2e overhead is negligible. In the case of Llama-3.1-8B + 96K context size + (K, L) = (10, 150) (the largest hash table we use in experiments ($\text{\textcolor{blue}{Sec 5.1}}$)), we present the time breakdown:
>
> | Prefilling |  Table Construction |  Total time with overlapping|
> | -----         | -----                         | -----|
> | 28s         |    20s                      | 29.5s |
>
> When the model finishes prefilling layer-i, we can (1) construct hash tables for layer-i on **CPU** and (2) prefill for layer-(i+1) at the same time on **GPU**.
>
> In fact, besides being able to do sampling in addition to search, the low construction time of hash tables is another advantage of LSH compared to other approximate nearest-neighbor search data structures, e.g., HNSW [1].
>
> **Factors:**  (1) LSH hyper-parameter (K, L). Larger (K, L) corresponds to longer construction time. (2) Sequence length. Longer sequences take longer to construct hash tables. (3) Hardware, e.g., the speed of CPUs.
>
> [1] Malkov, Yu A., and Dmitry A. Yashunin. "Efficient and robust approximate nearest neighbor search using hierarchical navigable small world graphs." IEEE transactions on pattern analysis and machine intelligence 42.4 (2018): 824-836.

---

> > ### Comment · Reviewer_Q8fb · 2024-11-29
> >
> > Thanks for authors reply. I would like to stay positive about this paper.

---

### Official Review · Reviewer_ePKx · 2024-11-03

**Soundness:** 3
**Presentation:** 2
**Contribution:** 3
**Rating:** 6
**Confidence:** 4

**Summary:**

This paper introduce a novel approach that leverages LSH sampling to approximate the oracle sampling. Empirical evaluation shows improvement over the baseline.

**Strengths:**

S1. The problem and the solution is well-motivated in the paper.
S2. The CPU-GPU co-design enables the storage of large LSH index table.
S3. The empirical results outperforms baselines.

**Weaknesses:**

Weak Points
----
W1. In the design of the proposed system, the authors claim that putting the retrieval stage in the CPU side would allow large hash tables. I wonder if moving the full system into GPU would reduce the latency when the GPU memory is sufficiently large to fit the hash table.

W2. The author discussed a few KV Cache reduction methods in Section 2. However, only quest is considered as the baselilne in the experiments. I would suggest the author to add a reasonable justification or add more baselines.

W3. Another direction of accelerating the inference is to quantize the model. How does the proposed method work on quantized LLM is not discussed.

W3. No code is provided. It might be hard for readers to reproduce the results.

Presentation
----
P1. In the abstract, without any notes, the author claims "achieve 110ms decoding latency on a single RTX 4090" while not actually running the code on RTX 4090. I believe this is a false claim without mentioning the simulation.
P2. Although it might be obvious for readers with retrieval and word extraction background, the acronym niah, cwe, and fwe and not explained before usage.
P3. The numbers in Figure 6 might be a bit outdated. In addition, the connection between CPU and GPU could be faster SXM.

**Questions:**

See weaknesses.

---

> ### Author Response · Authors · 2024-11-22
>
> Thank you very much for your thoughtful review. We are glad the reviewer found our work **well-motivated**, with **good system design** and **empirical results**. We have tried to address your questions carefully. We hope the reviewer will consider raising your score in light of our response.
>
> ## W1: Moving the full system into GPU would reduce the latency when the GPU memory is sufficiently large to fit the hash table.
>
> **Yes.** Moving the full system into GPU and applying GPU-friendly hashing functions can be more effective, which is a very promising future direction. **However**, the algorithm-system co-design and implementation are quite different if GPU memory is sufficient, such as H200/B200. Our system is a demonstration of a promising direction and is currently co-designed for $\text{\textcolor{blue}{low-cost LLM serving}}$ **(34B on 1xA100-80G, 8B on 1xRTX4090-24G, 13B on 1xL40-48G)** where high-end GPUs with large VRAM are usually **unavailable**.
>
> ## W2: I would suggest the author add a reasonable justification or add more baselines.
>
> We have added another baseline with dynamic KV cache sparsity, Loki [1]  in $\text{\textcolor{blue}{(Sec 5.1, Pg. 9 Table 3)}}$.
>
> |   Methods  |  Config  |   16K |   32K |  64K | 96K |  Avg | Total Cost |
> |    ------        | -----       |   -----  |  -----  |  -----  |  -----  | ----- |  -----         |
> | Llama-3.1|  Full      | 94.2     |91.5   | 86.1   | 83.0 |  88.7 |  1.0 |
> | MagicPIG  |(10,150)|  91.8   |88.9   | 84.8  |80.0  | 86.4  |  0.02|
> | MagicPIG  |(9,120)  | 93.4    |90.6   | 84.7   |81.5 | 87.6   |  0.04|
> | MagicPIG  |(8,75)    | 92.9   |90.2     | 84.9  |81.7 | 87.4   |  0.05|
> | Quest        |(16, 0.04)|86.3 |85.4      | 81.9  |74.9 | 82.1|  0.1|
> |Loki           |  (32, 0.03)| 80.0|63.6 | 61.9|34.7| 60.1 | 0.15|
>
> The configuration of Loki is low rank=32 and sparsity=3%.
> MagicPIG outperforms Loki in terms of accuracy vs. total cost.
>
> **Static KV cache** methods mentioned in our related work, like H2O[2] and StreamingLLM[3] suffer from severe accuracy loss in information retrieval tasks, as described in Quest paper [4] (Table 1), so we don’t provide additional experiments in our paper.
>
> [1] Singhania, Prajwal, et al. "Loki: Low-Rank Keys for Efficient Sparse Attention." arXiv preprint arXiv:2406.02542 (2024).
>
> [2] Zhang, Zhenyu, et al. "H2o: Heavy-hitter oracle for efficient generative inference of large language models." Advances in Neural Information Processing Systems 36 (2023): 34661-34710.
>
> [3] Xiao, Guangxuan, et al. "Efficient streaming language models with attention sinks." arXiv preprint arXiv:2309.17453 (2023).
>
> [4] Tang, Jiaming, et al. "Quest: Query-Aware Sparsity for Efficient Long-Context LLM Inference." arXiv preprint arXiv:2406.10774 (2024).
>
> ## W3: Compatibility with Quantization.
>
> We evaluate our methods with the Quanto package (4-bit quantization). Our proposed methods work well with 4-bit quantization across variable context lengths.
>
>
> |   Methods  |   16K |   32K |  64K | 96K | Total Cost |
> |    ------        |   -----  |  -----  |  -----  |  -----  | ----- |
> | Llama-3.1 + bfloat16 | 94.2     |91.5   | 86.1   | 83.0 | 1.0 |
> | Llama-3.1 + quanto  4bit |  94.2     |91.7   | 85.9   | 83.0 | 1.0 |
> | MagicPIG + quanto 4bit  |93.1    |89.8   | 84.9   |81.9 | 0.04|
>
> In the future version, we will evaluate more quantization methods (e.g., QServe [1], HQQ [2]).
>
> [1] Lin, Yujun, et al. "Qserve: W4a8kv4 quantization and system co-design for efficient llm serving." arXiv preprint arXiv:2405.04532 (2024).
>
> [2] https://huggingface.co/docs/transformers/main/en/quantization/hqq
>
> ## W4: No code is provided. It might be hard for readers to reproduce the results.
>
> We have uploaded the code.

---

> ### Author Response · Authors · 2024-11-22
>
> ## P1: Speed Claim.
> Thank you for raising the concern. We rented RTX 4090s to run Llama-3.1-8B-Instruct + 96K context experiments. RTX 4090 will slightly reduce the latency from 110ms (our result simulated with L40) to **107ms** as its bandwidth (1TB/s) is slightly faster than L40’s (864GB/s).
>
> ## P2: Acronyms.
>
> Thank you for raising the concern. Here, we explain the acronyms.
> niah: needle in a haystack; fwe: frequent work extraction; cwe: common word extraction
> We added notes to explain the acronyms on the paper in $\text{\textcolor{blue}{Section 3.1, Pg. 4}}$.
>
> ## P3: Connection between CPUs and GPUs.
>
> Our target setting is low-cost LLM serving. In this case, CPUs and GPUs are connected with PCIE4.0, which has a maximum bandwidth of 31.5GB/s.
>
> We also empirically measured the CPU-GPU bandwidth in the machine we used for experiments and got the following results,
>
> A100: 25-26GB/s, L40: 26-27GB/s.
>
> Both of them use PCIE4.0 to connect CPUs and GPUs.
>
> **We acknowledge that higher-end GPUs exist with faster data transfer speeds between CPUs and GPUs, which can potentially reduce latencies.**

---

> > ### Comment · Reviewer_ePKx · 2024-11-26
> >
> > Thank you for the responses. I would like to maintain my current score as it is.

---

### Official Review · Reviewer_UYfg · 2024-11-04

**Soundness:** 3
**Presentation:** 3
**Contribution:** 3
**Rating:** 8
**Confidence:** 4

**Summary:**

This paper proposes MAGICPIG for efficient attention score sampling to resolve the large KV-cache for LLM inference. The observation is that exact top-k attention sampling may not perform well. The proposal is to conduct sampling using locality sensitive hashing (LSH) and use importance sampling to obtain unbiased estimations. Empirical results show that MAGICPIG achieves high accuracy with low computation cost.

**Strengths:**

1.	The observation that top-k attention does not perform well is interesting.

2.	The idea of using LSH to conduct importance sampling is novel.

3.	MAGICPIG partitions the computation reasonably between GPU and CPU, i.e., the hashing (which involves matrix operations) is conducted on the GPU while attention computation is conducted on CPU.

**Weaknesses:**

1.	Lacks an intuitive explanation why LSH-based importance sampling works better than exact top-k attention. For the theorical view, I get it that importance sampling provides unbiased estimation while exact top-k attention does not. However, both importance sampling and top-k selects some attention scores to compute. Is it because (i) importance sampling select some scores that top-k will not select or (ii) once sampled, importance sampling assigns higher weights to scores with low sampling probabilities? It will be good if an ablation study can be conducted. For instance, if the case is (i), will it work if combine top-k sampling and sampling some random tokens (or some tokens at regular intervals of the sequence, for a good representation of the sequence)?

2.	The parameter configurations for LSH can be discussed, which involves the number of hash table (H), the number of hash functions for a hash table (L), the number of collisions for a token to be considered as a candidate for attention computation (T). Currently, T is fixed at 2. I understand that to sample a fixed number of attention scores, when H is increased, L should be reduced. We can also increase both H and L, but reduce T. Please provide some insights on how these parameters should be set.

3.	What are the current execution statistics of the system? When the CPU is computing the sampled attention scores, is the GPU idle? GPU or CPU has a longer running time? If we use a pipeline (e.g., by switching between two mini-batches) to overlap GPU and CPU computation, which one will be the straggler?

**Questions:**

See the weakness part

---

> ### Author Response · Authors · 2024-11-22
>
> Thank you very much for your insightful review. We are glad the reviewer found our work **interesting**, **novel** and the system design is **reasonable**. We have tried to address your questions carefully. We hope the reviewer will consider raising your score in light of our response.
>
> ## Q1:  Lacks an intuitive explanation why LSH-based importance sampling works better than exact top-k attention. For the theorical view, I get it that importance sampling provides unbiased estimation while exact top-k attention does not. However, both importance sampling and top-k selects some attention scores to compute. Is it because (i) importance sampling selects some scores that top-k will not select or (ii) once sampled, importance sampling assigns higher weights to scores with low sampling probabilities? It will be good if an ablation study can be conducted. For instance, if the case is (i), will it work if combine top-k sampling and sampling some random tokens (or some tokens at regular intervals of the sequence, for a good representation of the sequence)?
>
> Thank you for your insightful questions.
>
> $\text{\textcolor{blue}{Importance sampling assigns higher weights to scores with low sampling probabilities}}$ is the main reason sampling outperforms top-k; the re-weighting guarantees that the estimation is unbiased.
>
> The following experiments show that adding elements outside TopK cannot solve the problems.
> The **TopK + interval** uses half of the budget (i.e., attention computation cost) to select the KV cache with the TopK attention score, and the other half chooses tokens at **regular sequence intervals**.  We report the accuracy for niah_multikey3 and cwe (the same tasks as $\text{\textcolor{blue}{Figure 5c, Pg. 5}}$)
>
>
> |     Budget   |    TopK | TopK + interval |
> |      ------      |     ------ |  ------              |
> |      0.01        |     92/75.8       |        94/66.6          |
> |  0.004          |       90/60.8                  |         86/48.2       |
> |  0.002          |     86/48.2       |         88/37.2       |
>
> **Oracle sampling** yields an accuracy of **100/90.2** with a 0.002 budget.
>
> An intuitive understanding of how sampling can work better than TopK is that TopK only captures the **ranking** information when estimating attention output. In contrast, sampling considers the **entire data distribution** (i.e., the attention score after softmax).
>
> ### Intuitive Example
>
> We provide an intuitive example to explain why sampling can work better than TopK. Suppose a zoo has 100 animals in total: 10 elephants, 10 pigs, 10 tigers, and other 70 animals are all different kinds (each has only one). They eat 50lb, 20lb, 10lb, 1lb, 1lb, 1lb … of food every day. One wants to estimate the average weight of the food every animal eats in this zoo, which is $(50 \times10+20  \times10+ 10  \times 10+1  \times70  )/100 = 8.7$lb.
> - TopK (K=10) will select elephants, pigs, tigers and other 7 animals … and report the average to be $(50  \times10+20  \times10+ 10  \times10+7  \times1) / 37 = 22$ lb, which is biased.
> - Through the sampling process, which allows 10 trials. We sample with replacement from the probability distribution $[0.1, 0.1, 0.1, 0.01  \times 70]$ (constructed from the number of each animal, corresponding to the attention score vector).  For example, if the sampling trace is [elephant, pig, tigers, others $ \times$7], then we can give the estimation as $(50 + 20 + 10 + 7  \times 1) / 10 = 8.7$lb.
> - Even if there can be some variance among different sampling traces, theoretically, we can prove that sampling is unbiased and the std is 4.7lb, which is still better than the TopK estimation.
> - If we set the sampling budget as 20, then TopK will give the estimation as $(50 \times10+20 \times10+ 10  \times 10+17 \times1) / 47 = 17$ lb. Sampling will still give the unbiased estimation of 8.7lb, with std further reduced to 3.4lb.

---

> ### Author Response · Authors · 2024-11-22
> **Q2 (Part1)**
>
> ## Q2: The parameter configurations for LSH can be discussed, which involves the number of hash table (H), the number of hash functions for a hash table (L), the number of collisions for a token to be considered as a candidate for attention computation (T). Currently, T is fixed at 2. I understand that to sample a fixed number of attention scores, when H is increased, L should be reduced. We can also increase both H and L, but reduce T. Please provide some insights on how these parameters should be set.
>
> Thank you for raising this question. $\text{\textcolor{blue}{Finding the optimal (K, L) for high accuracy and efficiency is a long-standing problem in LSH}}$. We added a detailed discussion in "reply to all reviewers" about (K, L) and also added the discussion to  $\text{\textcolor{blue}{Appendix E.5 (Pg. 20)}}$.
>
> In MagicPIG, we **manually** set (K, L) based on the following ablations.
>
> Model: Llama-3.1-8K-Instruct; Task: RULER + 16k; Full model accuracy: **94.2**
>
> $\text{\textcolor{blue}{Exp1: Vary (K, L) and fix the attention computation cost/budget }}$
> | K       |  L |  Accuracy |  cost |
> |   ----- | -----| -----     | -----   |
> |  10    |  240|  94.2   | 4%|
> |   9     |  120| 92.8 | 4%|
> |   8     |  65  | 92.3 | 4%|
> |   7     |  35  |   88.5  |  4%|
>
> $\text{\textcolor{blue}{Exp2: Fix L as 120 and vary K (the budget will also vary) }}$
> | K       |  L |  ACC |  cost |
> |   ----- | -----| -----     | -----   |
> |  11   |   120| 60.2  |    0.5%|
> |  10    |  120| 87.3 | 1.2%|
> |   9     |  120|92.8 | 4%|
> |   8     |  120| 94.1 | 11%|
> |   7     |  120 | 94.3 | 27%|
>
> If we want the computation cost to be below 5% and L below 200 (to reduce memory overhead in CPU), then K=8-10 is a reasonable choice. Unlike K, L is not that sensitive. We select L **based on the following principle** after determining K: for larger K, we can allow the computation cost to be smaller since the sampling is more precise. This is why we choose to use (8, 75), (9, 120), and (10, 150).
>
> It’s worth pointing out that tuning (K, L) is a challenging problem in LSH , and we only give an example of practice in MagicPIG.
>
>
> We also provide some **notes** on how (K, L) influences LSH process,  the attention computation cost/budget, and how (K, L) is related to accuracy. A more detailed discussion is added in $\text{\textcolor{blue}{Appendix E (Pg. 18-20)}}$.
>
> **Note1: What (K, L) do with LSH.**  In each hash table, we use K hash functions to compute the hash code of $k$ and $q$.  In Simhash, i.e, the hashing we use in MagicPIG, the hash functions are random projections. With K random projections, we are able to partition the space (in our problem, the space is $R^d$) into $2^K$ subspace. If and only if $k$ and $q$ fall in the same subspace, we say $k$ and $q$ collide in this hash table.  We have L hash tables in total. In MagicPIG, if and only if $k$ and $q$ collide in at least two hash tables, $k$ is sampled/retrieved by $q$.   Intuitively,
>    - **if K is too small**, then we cannot partition the space well, we will sample too many ks, which might be actually far away from q (in the attention problem, this means their inner production is small), resulting in an increase in computation cost.
>    - On the other hand, **if K is too large**,  although the quality of sampled ks will be better, the collision probability in each table will be small thus the number of the sampled ks will be reduced. We need to increase L to make sure that at least a certain amount of keys are sampled and involved in the computation.  However, increasing (K, L) too much will bring more memory overhead on CPU DRAM, since we build L hash tables for each key-value head.
>   - Thus, (K, L) is important because it balances the computation cost, overhead and sampling quality (which determines the accuracy). Tuning (K, L) is necessary in LSH.

---

> ### Author Response · Authors · 2024-11-22
> **Q2 (Part2)**
>
> **Note2: (K, L) and computation cost/budget.** In summary, increasing K will make the budget smaller, and increasing L will increase the budget.
>   - $\text{\textcolor{blue}{(Theoretically)}}$  As introduced in $\text{\textcolor{blue}{Section 4.3 (Pg. 7)}}$, in our approach, the key $k_i$ is sampled only if at least two hash tables exist where $k_i$ shares the hash value with query $q$. With the assumption that $k_i$ is well-distributed (In each hash table out of L, each hash value corresponds to roughly the same number of $k_i$s), the ratio of retrieved $k_i$s can be estimated with
> $\mathcal{B} / n = 1 - (1 - 0.5^K)^L - L \times 0.5^K (1 - 0.5^K)^{(L-1)} $, where $n$ is the context length, here, we estimate the collision probability of $k_i$ and $q$ in a single hash table as $0.5^K$.
>
>   - $\text{\textcolor{blue}{(Empirically)}}$ The ratio of retrieved keys and values $\mathcal{B} / n$ might differ from the above estimation since the data is not perfectly distributed.  In our experiments, after fixing (K, L), we **empirically** measure the number of keys and values accessed each time and report their averages. We present the empirically measured budget below,
>
> | K / L | 75 | 100 | 120 | 150 | 200 | 300|
>  | ---|---|---|---|---|---|---|
> |  7      | 14% | 21%| 27% |35%| 48% | 66%|
> |  8      |  5%|  8% | 11%| 15% | 22% | 36%|
> |  9      |  1.6% | 2.7%| 4% | 5.4%| 8.5% | 15.44%|
>   |  10    | 0.5% |  0.9% | 1.2% | 2% | 3%| 6%|
>  |   11   | 0.15% | 0.3%| 0.5%|  0.6% | 1%| 2%|
>
> **Note3: (K, L) and accuracy.**  There is no simple relationship between (K, L) and downstream accuracy since (K, L) not only influences sampling quality but also influences the computation budget. One safe way to discuss the relation between (K, L) and accuracy is: Fixing the computation budget, larger (K, L) will potentially produce higher accuracy, since the sampling quality is higher.  Our experimental results show that,
>
> - $\text{\textcolor{blue}{Increasing (K, L) can significantly improve accuracy in relatively longer contexts}}$
>
> Model: MegaBeam-7B-512K
> |   Methods  |  Config  |  16K |128K | 256K | Cost |
> |    ------        | -----       | -----|-----  |  -----  |  -----  |
> | |  Full      |  91.7 |83.7   | 82.5|   1.0 |
> | MagicPIG  |(10,150) |  89.8 | 80.7|  79.0|   0.02|
> | MagicPIG  |(11,300)  |  90.6    |83.3|  81.9|   0.02|
>
> - $\text{\textcolor{blue}{Same set of (K, L) can generalize to larger LLMs}}$
>
> |   Models / Config  |  Full | (10,135) |(10, 150) | (9, 110) | (9, 120) |
> |    ------        | -----       | -----|-----  |  -----  |  -----  |
> |    Llama3.1-8B-Instruct   |  86.1      |  83.6 |84.8   | 84.7|   84.7 |
> |    Llama3.1-70B-Instruct   |  89.1      |  86.7 |88.2   | 88.4|  89.1 |

---

> ### Author Response · Authors · 2024-11-22
>
> ## Q3 What are the current execution statistics of the system? When the CPU is computing the sampled attention scores, is the GPU idle? GPU or CPU has a longer running time? If we use a pipeline (e.g., by switching between two mini-batches) to overlap GPU and CPU computation, which one will be the straggler?
>
> We first present the breakdown of our system's execution time above.
>
> $\text{\textcolor{blue}{Model: Llama-3.1-8B-Instruct; Context Length: 96K}}$
> |   Batch size  |  CPU  |  Data Transfer | GPU |
> |    ------        | -----       |   -----  |  -----  |
> | 1  |  64ms  | 6ms |  40ms |
> | 4  |  128ms| 6ms |    43ms|
>
> $\text{\textcolor{blue}{Model: CodeLlama-34B; Context Length: 16K }}$
> |  Batch size  |  CPU  |  Data Transfer | GPU |
> |    ------        | -----       |   -----  |  -----  |
> |       2           |       31ms      |  8ms |   56ms  |
> |      18         |        190ms    |    10ms      |   66ms |
>
> To answer your question:
> - GPU **is idle** when the CPU is computing the sampled attention scores. But with a CPU-GPU pipeline (under development), we can further boost the throughput of MagicPIG.
> - The running time of CPUs and GPUs depends on the **workload**. For example, in the Llama-3.1-8B-Instruct + 96K context size case, the GPU part is lightweight, and the CPU's KV cache will dominate, making the CPU part a bottleneck.  **However**, in the CodeLlama-34B + 16K context size case,  model weights (involved in GPU computation) are larger than the KV cache until the batch size is very large.
>
> Regarding your question, we have an ongoing system optimization plan if you are interested.
>
> **Additional future plan**: It’s worth pointing out that our current system implementation still has a lot of room to optimize. For example,
> - CPU-GPU pipeline. As you mentioned and discussed in FastDecode [1], can further boost our system throughput.
> - Our current implementation uses one open-mp thread to process one attention head. When the batch size is small, some CPU cores are idle during decoding, so thread scheduling is necessary in this case. Our preliminary result shows that, by simply splitting the workload in one attention head to multiple cores, we can reduce the latency $\text{\textcolor{blue}{from 110 ms to around 75 ms }}$ for Llama-3.1-8B + 96K context.
>
> [1] He, Jiaao, and Jidong Zhai. "FastDecode: High-Throughput GPU-Efficient LLM Serving using Heterogeneous Pipelines." arXiv preprint arXiv:2403.11421 (2024).

---

> > ### Comment · Reviewer_UYfg · 2024-11-25
> > **Afte author response**
> >
> > Thank you for the reponse. It address my concerns. It will improve the paper if some contents (e.g., the intuition why top-k is not good and the parameter configuration) can be added to the paper.

---

> > > ### Author Response · Authors · 2024-11-26
> > >
> > > Thank you for your valuable feedback! We updated Appendix E and Appendix F to provide a detailed discussion on parameter configurations and why sampling can outperform top-k. We greatly appreciate your constructive comments!

---

### Official Review · Reviewer_KBnM · 2024-11-04

**Soundness:** 3
**Presentation:** 4
**Contribution:** 3
**Rating:** 8
**Confidence:** 4

**Summary:**

This paper studies the optimization of long-context LLM inference. Unlike most existing approaches that mainly adopt TopK selection for attention calculation, this paper presents a novel method based on importance sampling, where SimHash is used for estimation. Experiments on a set of benchmarks demonstrate the effectiveness of the proposed method and its superiority over a state-of-the-art TopK selection approach.

**Strengths:**

S1. This paper studies from a new perspective of estimating attention scores, in contrast to TopK selection that has been widely targeted in existing works. The proposed approach showcases its potential in dealing with the non-sparse case of attention.

S2. Experiments are highly promising, outperforming Quest in both accuracy and efficiency.

S3. System design is discussed, with the jobs of CPU and GPU clearly depicted in the figure.

**Weaknesses:**

W1. The context length seems to be short in the evaluation.

W2. Some parameter evaluations are missing in the experiments.

W3. Only Llama series models are evaluated.

**Questions:**

Q1. In Figure 3, did you mean even the exact TopK selection yields a higher relative error than oracle sampling? For oracle sampling, I suppose you estimate the weight of each value vector in the attention. As such, a better result can be obtained, especially for the case when attention is not sparse, where TopK selection treats all non-TopK values as zero-weights.

Q2. For LSH, why SimHash was chosen? The method proposed by Andoni et al. (Practical and optimal LSH for angular distance, NeurIPS 2015) is a better approach than SimHash and has been used in Reformer.

Q3. How does the budget B relate to K and L? For each hash probe out of L, there could be multiple k_i's having a hash collision with q. I suppose the number of retrieved k_i's in L hash probes should be reflected to the budget.

Q4. Following Q3, hash collision could be a problem when the context goes long. In this case, K and L can be adjusted to strike a balance, but it is unclear how they are affected by the context length (the context used in the paper seems to be short, see Q5).

Q5. What is the maximum context length used in the experiments? It seems to be 96K. It is encouraged to see what if the context goes longer, e.g., 1M, which has been evaluated in some TopK-based approaches such as InfLLM.

Q6. Despite evaluating the importance of centering, Figure 8(a) can be seen also as an evaluation of the impact of L. However, I didn't find the evaluation of K. I wonder how K = 8-10 was determined in the experiment.

Q7. On LongBench and RULER, the performance is even higher when a smaller set of (K, L) is used, e.g. (8, 75) and (9, 120), in comparison to (10, 150). Why?

**Details Of Ethics Concerns:**

As a study on LLM core technology, this paper has nothing flagged for ethics review.

---

> ### Author Response · Authors · 2024-11-22
>
> Thank you very much for your thoughtful review and constructive suggestions. We are glad the reviewer found our work **novel** and **empirically effective**. We have tried to address your questions carefully. We hope the reviewer will consider raising your score in light of our response.
>
>
> ## W1& W3 & Q5: “The context length seems short in the evaluation”. “Only Llama series models are evaluated”. “It is encouraged to see what if the context goes longer, e.g., 1M, which has been evaluated in some TopK-based approaches such as InfLLM.”
>
> In the revised paper, we have included evaluation on long-context benchmarks, including MegaBeam-Mistral-7B-512K [1] and Llama-3-8B-Prolong-512K-Instruct [2] $\text{\textcolor{blue}{(Appendix D.1, Pg. 17)}}$, and demonstrated MagicPIG can maintain high accuracy when scaling to longer contexts and can generalize to models beyond the Llama family.  (Currently, our evaluation is up to 256K due to the time limit for experiments. We will include the additional models and results in the next revision.)
>
> MegaBeam-Mistral-7B-512K
> |   Methods  |  Config  |   16K |   32K |  64K | 96K | 128K | 256K |  Avg | Total Cost |
> |    ------        | -----       |   -----  |  -----  |  -----  |  -----  | ----- | -----  |  -----    |  -----         |
> |      Full        |   | 91.7   |  88.1 | 83.5  |83.7   | 83.5|  82.5|  85.5   |  1.0 |
> | MagicPIG  |(10,150)|  89.8   | 86.5  | 81.7  |80.7   | 81.6|  79.0|  83.2   |  0.02|
> | MagicPIG  |(9,120)  | 90.7    | 88.5 |  82.9  | 82.4  | 82.3 |80.1 |  **84.5**    |  0.04|
> | MagicPIG  |(8,75)    | 90.6    | 86.4 |  82.8  | 81.6  | 82.3 | 80.8| 84.1    |  0.05|
> | Quest        |(16, 0.04)| 83.3  | 83.2 | 79.3  |  78.6 |  78.5 |  78.5 | 80.2  |  0.10|
>
> Llama-3-8B-Prolong-512K-Instruct
> |   Methods  |  Config  |   16K |   32K |  64K | 96K | 128K | 256K |  Avg | Total Cost |
> |    ------        | -----       |   -----  |  -----  |  -----  |  -----  | ----- | -----  |  -----    |  -----         |
> |      Full      |   |93.5   | 90.8 | 85.1  | 83.5  |  81.7 | 78.4  |  85.5 |   1.0|
> | MagicPIG  |(10,150)| 88.0    |86.4  | 81.3   |78.8  |77.3   |71.1   | 80.5  | 0.02|
> | MagicPIG  |(10,170)| 89.0    |88.7  | 82.8   |80.0  |77.7   |73.7   | 82.0  | 0.025|
> | MagicPIG  |(9,120)  | 91.4    |88.2 |  82.4  | 80.4 | 79.2  |75.2  | **82.8**  |  0.04|
> | MagicPIG  |(8,75)    | 91.4    |88.6 |  83.1   |80.5 | 79.1 | 73.9   | **82.8** | 0.05|
> | Quest        |(16, 0.04)| 84.9  |83.7 | 78.7   |78.6  | 76.3 |72.1    | 79.2 | 0.10|
>
> Thanks for pointing out several missing related works! We have added the discussion on prior work targeting extremely long-context scenarios via context extrapolation, such as StreamingLLM [3] and InfLLM [4], which can extend to several millions of contexts, in the revised paper $\text{\textcolor{blue}{(Sec 2.2, Pg. 4)}}$.
>
> [1]  https://huggingface.co/aws-prototyping/MegaBeam-Mistral-7B-512k
>
> [2]  Gao, Tianyu, et al. "How to train long-context language models (effectively)." arXiv preprint arXiv:2410.02660 (2024).
>
> [3] Xiao, Guangxuan, et al. "Efficient streaming language models with attention sinks." arXiv preprint arXiv:2309.17453 (2023).
>
> [4] Xiao, Chaojun, et al. "Infllm: Unveiling the intrinsic capacity of llms for understanding extremely long sequences with training-free memory." arXiv preprint arXiv:2402.04617 (2024).

---

> ### Author Response · Authors · 2024-11-22
>
> ## Q1: Did you mean even the exact TopK selection yields a higher relative error than Oracle sampling?
>
> Your understanding is **correct**, as shown in $\text{\textcolor{blue}{Figure 5(a)(b), Pg. 5}}$. Oracle sampling can give an unbiased estimation according to the attention scores (after softmax), thus considering every value, while TopK is a biased estimation. As a result, Oracle sampling can produce better estimation, especially when the attention score is not that sparse.
>
> ### Intuitive Example
> We provide an intuitive example to explain why sampling can work better than TopK. Suppose a zoo has 100 animals in total: 10 elephants, 10 pigs, 10 tigers, and other 70 animals are all different kinds (each has only one). They eat 50lb, 20lb, 10lb, 1lb, 1lb,  … of food every day. One wants to estimate the average weight of the food every animal eats in this zoo, which is $(50 \times 10+20 \times10+ 10 \times 10+1 \times70  )/100 = 8.7$lb.
> - TopK (K=10) will select elephants, pigs, tigers and other 7 animals … and report the average to be $(50 \times 10+20 \times10+ 10 \times 10+7 \times1) / 37 = 22$ lb, which is biased.
> - Through the sampling process, which allows 10 trials. We sample with replacement from the probability distribution $[0.1, 0.1, 0.1, 0.01 \times 70]$ (constructed from the number of each animal).  For example, if the sampling trace is [elephant, pig, tigers, others $\times$ 7], then we can give the estimation as $(50 + 20 + 10 + 7 \times 1) / 10 = 8.7$lb.
> - Even if there can be some variance among different sampling traces, theoretically, we can prove that sampling is unbiased and the std is 4.7lb, which is still better than the TopK estimation.  If we set the sampling budget as 20, then TopK will give the estimation as $(50 \times 10+20 \times10+ 10 \times 10+17 \times1) / 47 = 17$ lb. Sampling will still give the unbiased estimation of 8.7lb, with std further reduced to 3.4lb.
>
>
> ## Q2: For LSH, why SimHash was chosen?
>
> We chose SimHash mainly for its **simplicity**. In SimHash, both hash codes and sampling probability can be easily computed by matrix multiplication with **simple close form** (corresponding to cosine distance), which is easy to implement in current LLM inference systems. More advanced Hashing (including cross-polytope and other data-dependent ones [2]) can benefit
> - more precise retrieval and estimation
> - saving space in CPU DRAM for maintaining hash tables.
>
> However, the sampling probability is not as easy to derive because they don’t have a simple close form.  How to approximate and obtain sampling probability of other advanced LSH algorithms is an important part of our future work.
>
> [1] Kitaev, Nikita, Łukasz Kaiser, and Anselm Levskaya. "Reformer: The efficient transformer." arXiv preprint arXiv:2001.04451 (2020).
>
> [2] Andoni, Alexandr, and Ilya Razenshteyn. "Optimal data-dependent hashing for approximate near neighbors." Proceedings of the forty-seventh annual ACM symposium on Theory of computing. 2015.
>
> ## Q3: How does the budget B relate to K and L?
>
> We added a detailed discussion in "reply to all reviewers" about (K, L) and also added the discussion to $\text{\textcolor{blue}{Appendix E (Pg. 18)}}$.
>
> **In summary, increasing K will make the budget smaller, and increasing L will increase the budget.**
>
> $\text{\textcolor{blue}{(Theoretically)}}$  As introduced in $\text{\textcolor{blue}{Section 4.3 (Pg. 7)}}$, in our approach, the key $k_i$ is sampled only if at least two hash tables exist where $k_i$ shares the hash value with query $q$. With the assumption that $k_i$ is well-distributed (In each hash table out of L, each hash value corresponds to roughly the same number of $k_i$s), the ratio of retrieved $k_i$s can be estimated with
>
> $\mathcal{B} / n = 1 - (1 - 0.5^K)^L - L \times 0.5^K (1 - 0.5^K)^{(L-1)} $
>
> where $n$ is the context length, here, we estimate the collision probability of $k_i$ and $q$ in a single hash table as $0.5^K$.
>
> $\text{\textcolor{blue}{(Empirically)}}$ The ratio of retrieved keys and values $\mathcal{B} / n$ might differ from the above estimation since the data is not perfectly distributed.  We present the empirically measured budget below,
>
> | K / L | 75 | 100 | 120 | 150 | 200 | 300|
>  | ---|---|---|---|---|---|---|
> |  7      | 14% | 21%| 27% |35%| 48% | 66%|
> |  8      |  5%|  8% | 11%| 15% | 22% | 36%|
> |  9      |  1.6% | 2.7%| 4% | 5.4%| 8.5% | 15.44%|
>   |  10    | 0.5% |  0.9% | 1.2% | 2% | 3%| 6%|
>  |   11   | 0.15% | 0.3%| 0.5%|  0.6% | 1%| 2%|
>
> In our experiments, after fixing (K, L), we **empirically** measure the number of keys and values accessed each time and report their averages.

---

> ### Author Response · Authors · 2024-11-22
>
> ## Q4: A hash collision could be a problem when the context goes long. In this case, K and L can be adjusted to strike a balance, but it is unclear how they are affected by the context length.
>
> You are **correct** that from a sampling perspective, fixing the computation budget B, (K, L) needs to be increased (for more accurate retrieval) when context size increases [1]. Adjusting (K, L) helps improve performance.  We present results with 128K and 256K contexts to support this point. We add this discussion to  $\text{\textcolor{blue}{Appendix E.4 (Pg. 20, Table 8)}}$.
>
> |   Methods  |  Config  |  16K |128K | 256K | Total Cost |
> |    ------        | -----       | -----|-----  |  -----  |  -----  |
> | MegaBeam|  Full      |  91.7 |83.7   | 82.5|   1.0 |
> | MagicPIG  |(10,150) |  89.8 | 80.7|  79.0|   0.02|
> | MagicPIG  |(11,300)  |  90.6    |83.3|  81.9|   0.02|
>
> In fact, finding the optimal (K, L) according to data size has been a long-standing problem in LSH[2].
>
> [1] https://en.wikipedia.org/wiki/Locality-sensitive_hashing
>
> [2] Qin Lv, William Josephson, Zhe Wang, Moses Charikar, and Kai Li. 2017. Intelligent probing for locality sensitive hashing: multi-probe LSH and beyond. Proc. VLDB Endow. 10, 12 (August 2017), 2021–2024. https://doi.org/10.14778/3137765.3137836
>
> ## Q6 Despite evaluating the importance of centering, Figure 8(a) can be seen also as an evaluation of the impact of L. However, I didn't find the evaluation of K. I wonder how K = 8-10 was determined in the experiment.
>
> Thank you for raising this question.
>
> Currently, K=8-10 is manually determined and found to be effective across various models and tasks.
>
> K is a more sensitive hyperparameter than L.  A slight change of K can drastically influence the number of retrieved items (i.e., budget) and quality. So, we acknowledge that how to determine K is a critical problem in LSH and is worth doing ablations and tuning.  We added a detailed discussion in "reply to all reviewers" about (K, L) and also added the discussion to $\text{\textcolor{blue}{Appendix E.5 (Pg. 20)}}$
>
> Here, we present the two ablations of hyper-parameter K to show how we determine it in practice.
>
> Model: Llama-3.1-8K-Instruct; Task: RULER + 16k; Full model accuracy: **94.2**
>
> - $\text{\textcolor{blue}{Exp1: Vary L and fix the computation cost/budget}}$
>
> | K       |  L |  Accuracy |  cost |
> |   ----- | -----| -----     | -----   |
> |  10    |  240|  94.2   | 4%|
> |   9     |  120| 92.8 | 4%|
> |   8     |  65  | 92.3 | 4%|
> |   7     |  35  |   88.5  |  4%|
>
>
>
> - $\text{\textcolor{blue}{Exp2: Fix L as 120 and vary K (the budget will also vary)}}$
>
> | K       |  L |  ACC |  cost |
> |   ----- | -----| -----     | -----   |
> |  11   |   120| 60.2  |    0.5%|
> |  10    |  120| 87.3 | 1.2%|
> |   9     |  120|92.8 | 4%|
> |   8     |  120| 94.1 | 11%|
> |   7     |  120 | 94.3 | 27%|
>
>
> If we want the computation cost below 5% and L below 200 (to reduce memory overhead in CPU), then K=8-10 is a reasonable choice. Unlike K, L is not that sensitive. We select L based on the following principle: after determining K, for larger K, we can allow the computation cost to be smaller since the sampling is more precise. This is why we choose to use (8, 75), (9, 120), and (10, 150).
>
>
> ## Q7:  On LongBench and RULER, the performance is even higher when a smaller set of (K, L) is used, e.g. (8, 75) and (9, 120), in comparison to (10, 150). Why?
>
> Although the LSH parameters are smaller (which means the quality of retrieved items is lower), the retrieved number of keys and values (i.e., **budget**) of (8, 75) and (9, 120) are **larger** than (10, 150).
>
> As we described in “$\text{Cost}_2$” in  $\text{\textcolor{blue}{Table 1/2/3, Pg. 9}}$, the corresponding budget of each LSH parameter are:
> |(K, L)  |   Budget |
> | --- | ---|
> |(10, 150)| 1.5% - 2%|
> |(9, 120) |3.7% - 4.1%|
> |(8, 75) |4.9% - 5.1%|
>
>
> This difference causes (8,75) and (9, 120) to perform better on specific tasks depending on the data (KV cache) distribution.  Finding the optimal (K, L) is a long-standing problem in LSH. More details are discussed in  $\text{\textcolor{blue}{Appendix E.3 and Appendix E.4 (Pg. 19-20)}}$.

---

> > ### Comment · Reviewer_KBnM · 2024-11-24
> > **Re: Author Response**
> >
> > Thanks for the authors' response. They addressed most of my concerns.
> >
> > I have two additional questions:
> >
> > Q8. From the paper, it is unclear how prefilling is processed in the proposed method. I suppose you also use the LSH-based attention (instead of full attention) in this stage, right? It seems the reported throughput only considers time per output token (TPOT). For long context, prefilling time also becomes an issue. So I wonder how your prefilling compares to the baseline in terms of efficiency, e.g., by time to first token (TTFT).
> >
> > Q9. May I have the result for throughput comparison against the baseline for the 256K context (and possibly break down into TTFT and TPOT if you also optimize prefilling)? I didn't find it in the revised paper. Only costs were reported. I think throughput is a more important measure.

---

> ### Author Response · Authors · 2024-11-25
>
> Thank you for your response!
>
> ## Q8
>
> Thank you for raising this question. Currently, our algorithm is **not targeted** on the prefilling stage in this work, so the prefilling throughput and TTFT remain the same as full attention baselines (also the same as Quest [1], Loki [2]).
>
> We acknowledge that prefilling time also becomes an issue in long context tasks, and extending MagicPIG to the prefilling stage is our next step. In the prefilling stage, we can also do sampling based on LSH. After obtaining the sampled indices, there are several existing libraries to perform the sparse attention efficiently (given the sparsity is <= 2%), such as **Flashinfer (masked Sparse Attention [3])**, **FlashMask [4]**, and the variants on CPU, such as **MKL sparse operators [5]**.
>
> Besides, for the current implementation, there are several ways to reduce the prefilling time that can be combined with our system. For example, **Prefilling / Decoding disaggregation** (Splitwise [6], Distserve [7], Mooncake [8]) (allocating different computation resources to prefilling ) and **Share-Prefix attention** (RadixAttention [9] and Chunkattention [10], also Sarathi [11][12]) (reducing recomputation).
>
> We added this in "Limitations and Future work."
>
>
> [1] Tang, Jiaming, et al. "Quest: Query-Aware Sparsity for Efficient Long-Context LLM Inference." arXiv preprint arXiv:2406.10774 (2024).
>
> [2] Singhania, Prajwal, et al. "Loki: Low-Rank Keys for Efficient Sparse Attention." arXiv preprint arXiv:2406.02542 (2024).
>
> [3] https://docs.flashinfer.ai/index.html
>
> [4] Wang, Guoxia, et al. "FlashMask: Efficient and Rich Mask Extension of FlashAttention." arXiv preprint arXiv:2410.01359 (2024).
>
> [5] https://www.intel.com/content/www/us/en/docs/onemkl/developer-reference-c/2024-2/sparse-blas-level-1-routines.html
>
> [6] Patel, Pratyush, et al. "Splitwise: Efficient generative llm inference using phase splitting." 2024 ACM/IEEE 51st Annual International Symposium on Computer Architecture (ISCA). IEEE, 2024.
>
> [7] Zhong, Yinmin, et al. "Distserve: Disaggregating prefill and decoding for goodput-optimized large language model serving." arXiv preprint arXiv:2401.09670 (2024).
>
> [8] Qin, Ruoyu, et al. "Mooncake: A kvcache-centric disaggregated architecture for llm serving." arXiv preprint arXiv:2407.00079 (2024).
>
> [9] Zheng, Lianmin, et al. "Efficiently programming large language models using sglang." arXiv e-prints (2023): arXiv-2312.
>
> [10] Ye, Lu, et al. "Chunkattention: Efficient self-attention with prefix-aware kv cache and two-phase partition." arXiv preprint arXiv:2402.15220 (2024).
>
> [11] Agrawal, Amey, et al. "Sarathi: Efficient llm inference by piggybacking decodes with chunked prefills." arXiv preprint arXiv:2308.16369 (2023).
>
> [12] Agrawal, Amey, et al. "Taming throughput-latency tradeoff in llm inference with sarathi-serve." arXiv preprint arXiv:2403.02310 (2024).
>
> ## Q9
>
> Here we extend the efficiency experiments from 96K to 256K using Llama3-8B-Prolong-512K-Instruct.
>
> $\text{\textcolor{blue}{Prefill}}$
>
> - **Baseline:** 1700 tokens/s.
>
> - **MagicPIG:** 1700 tokens/s.
>
> $\text{\textcolor{blue}{Decode}}$
>
> - **Baseline:** (Cannot fit in GPUs, even if batch size = 1) Maximum throughput: $\text{\textcolor{blue}{1.4 Tokens/sec}}$
> - **MagicPIG:** (24GB VRAM limits)
>
> | Batch Size |  Throughput (Tokens/sec)|
> | ----| ----|
> |  1  |  4.69 |
> |  2  |  7.29 |
> |  3  |  $\text{\textcolor{blue}{7.69}}$ |
> |  4   | OOM  |
>
> PS: For the CPU execution part, we have not implemented the thread scheduler. We use one open-mp thread to compute one attention head. When the batch size is small, some CPU cores might be idle, resulting in a relatively long execution time compared to a large batch size. Our preliminary result shows that, by simply splitting the workload in one attention head to multiple cores, we can increase the throughput $\text{\textcolor{blue}{from 4.69 Tokens/sec to 6.63 Tokens/sec}}$ for Llama-3-8B-Prolong + 256K context with $\text{\textcolor{blue}{batch size = 1}}$.
>
>
> Let us know if you have additional questions.  Thanks for your engagement!

---

> > ### Comment · Reviewer_KBnM · 2024-11-25
> > **Re: Author Response**
> >
> > Thanks for the authors' effort! I will raise my score accordingly.

---

> > > ### Author Response · Authors · 2024-11-26
> > >
> > > Thank you for your valuable feedback!

---

### Author Response · Authors · 2024-11-21
**Response to all reviewers**

We thank all the reviewers [**R1 (KBnM), R2 (Uyfg), R3 (ePKx), R4 (Q8fb), R5 (qU5o)**] for their thoughtful and highly supportive feedback! We were glad that the reviewers found the work **novel and promising [R1, R2, R3, R5]**, with **empirically solid results [R1, R2, R3, R4, R5]** and **theoretical guarantees [R4]** felt the founding is **interesting [R2]**, and believed our approach is **pivotal [R4]** for scaling LLMs in resource-constrained settings.

We have updated the paper to incorporate constructive suggestions, as shown in the revision. We summarize the major changes:
- **Evaluations on longer contexts and various models [R1]**:  We extended the maximum evaluated context lengths to 256K and evaluated two additional models, i.e., MegaBeam-Mistral-7B-512K and Llama3-Prolong-512K, showing 3.6% to 4.3% absolute accuracy improvement on average, compared to SOTA (i.e., Quest), with significantly smaller computation cost. We added the results in $\text{\textcolor{blue}{Appendix D.1 (Pg. 18, Table 4)}}$.
- **More baselines [R3]**: We added another dynamic KV cache sparsity algorithm, Loki, to our comparison, showing that our methods can significantly outperform Loki by over 25% on average. The results are added in $\text{\textcolor{blue}{Section 5.1 (Pg. 9, Table 3)}}$.
- **Analysis of LSH overhead [R4]**:  We analyzed the memory and computation overhead introduced by LSH sampling. The analysis is presented in $\text{\textcolor{blue}{Appendix E.2 (Pg. 19, Table 6)}}$.
- **LSH hyper-parameters (K, L) configuration [R1, R2, R4, R5]**: We analyzed the questions of how (K, L) is related to computation cost, accuracy, and how to select (K, L) with detailed ablation study presented in $\text{\textcolor{blue}{Appendix E (Pg. 18 - 20)}}$. We also add a note discussing LSH hyper-parameters as a reference to reviewers.
- **Impact of model sizes [R4]**:  We evaluated our approach with Llama-3.1-70B-Instruct, showing that the same set of LSH hyper-parameters (K, L) work well when scaling to larger models.  We added the results in $\text{\textcolor{blue}{Appendix D.2 (Pg. 18, Table 5)}}$.  We also present the size and overhead of hash tables for different sizes of models in $\text{\textcolor{blue}{Appendix E.2 (Pg. 19, Table 6)}}$.

---

> ### Author Response · Authors · 2024-11-21
> **Notes on LSH hyper-parameters**
>
> ## What (K, L) do in LSH
>
> In each hash table, we use K hash functions to compute the hash code of $k$ and $q$.  In Simhash, i.e., the hashing we use in MagicPIG, the hash functions are random projections. With K random projections, we can partition the space (in our problem, the space is $R^d$) into $2^K$ subspaces. If and only if $k$ and $q$ fall in the same subspace, we say $k$ and $q$ collide in this hash table.  We have L hash tables in total. In MagicPIG, if and only if $k$ and $q$ collide in at least two hash tables, $k$ is sampled/retrieved by $q$.  Intuitively,
> - if K is too small, then we cannot partition the space well; we will sample too many $k$s, which might be actually far away from $q$ (in the attention problem, this means their inner production is small), resulting in an increase in computation cost.
> - On the other hand, if K is too large, although the quality of sampled $k$s will be better, the collision probability in each table will be small, thus reducing the number of sampled $k$s. We need to increase L to ensure that a certain number of keys are sampled and involved in the computation. However, increasing (K, L) too much will bring more memory overhead on CPU DRAM since we build L hash tables for each key-value head.
>
> Thus, (K, L) is important because it balances computation cost, overhead, and sampling quality (which influences accuracy). Tuning (K, L) is necessary in LSH.
>
>
> ##  (K, L) and memory overhead
>
> (K, L) will change the memory occupied by hash tables on the CPU.  We give examples here.
>
> |Models  | (K, L) | Context length | Size of Projectors | Size of Hash tables |
> |---|---|---|---|---|
> |Llama-3.1-8B-Instruct | (10, 150) | 96K | 384KB | 14GB|
> |Llama-3.1-8B-Instruct | (11, 300) | 96K | 825KB | 28GB|
> |Llama-3-8B-512K | (10, 150) | 256K | 384KB | 37GB|
> |Llama-3-8B-512K | (11, 300) | 256K | 825KB | 74GB|
> |Llama-3.1-70B-Instruct | (10, 150) | 96K | 384KB | 70GB|
>
> The enlarged size of hash tables prevents us from always increasing (K, L) to get more accurate results.
>
> As shown in the table above, under the same (K, L), the memory overhead of hash tables grows linearly with **context length** and the total number of key-value heads in models (which is determined by **model sizes**).
>
> ## (K, L) and computation cost/budget.
>
> Here the cost/budget refers to the $Cost_{2}$ in $\text{\textcolor{blue}{Table 1/2/3, Pg. 9}}$.
> In summary, increasing K will make the budget smaller, and increasing L will increase the budget.
>
> $\text{\textcolor{blue}{(Theoretically)}}$  As introduced in $\text{\textcolor{blue}{Section 4.3 (Pg. 7)}}$, in our approach, the key $k_i$ is sampled only if at least two hash tables exist where $k_i$ shares the hash value with query $q$. With the assumption that $k_i$ is well-distributed (In each hash table out of L, each hash value corresponds to roughly the same number of $k_i$s), the ratio of retrieved $k_i$s can be estimated with
>
> $\mathcal{B} / n = 1 - (1 - 0.5^K)^L - L \times 0.5^K (1 - 0.5^K)^{(L-1)} $
>
> where $n$ is the context length, here, we estimate the collision probability of $k_i$ and $q$ in a single hash table as $0.5^K$.
>
> $\text{\textcolor{blue}{(Empirically)}}$ The ratio of retrieved keys and values $\mathcal{B} / n$ might differ from the above estimation since the data is not perfectly distributed.  We present the empirically measured budget below,
>
> | K / L | 75 | 100 | 120 | 150 | 200 | 300|
>  | ---|---|---|---|---|---|---|
> |  7      | 14% | 21%| 27% |35%| 48% | 66%|
> |  8      |  5%|  8% | 11%| 15% | 22% | 36%|
> |  9      |  1.6% | 2.7%| 4% | 5.4%| 8.5% | 15.44%|
>   |  10    | 0.5% |  0.9% | 1.2% | 2% | 3%| 6%|
>  |   11   | 0.15% | 0.3%| 0.5%|  0.6% | 1%| 2%|
>
> In our experiments, after fixing (K, L), we **empirically** measure the number of keys and values accessed each time and report their averages.
>
>
> ## (K, L) and accuracy
>
> There is no simple relationship between (K, L) and accuracy since (K, L) not only influences sampling quality but also the computation budget. One safe way to discuss the relation between (K, L) and accuracy is: Fixing the computation budget, larger (K, L) will potentially achieve higher accuracy, since the sampling quality is higher. Our experimental results show that,
>
> - Increasing (K, L) can significantly improve accuracy in relatively longer contexts.
> Model: MegaBeam-7B-512K
> |   Methods  |  Config  |  16K |128K | 256K | Cost |
> |    ------        | -----       | -----|-----  |  -----  |  -----  |
> | |  Full      |  91.7 |83.7   | 82.5|   1.0 |
> | MagicPIG  |(10,150) |  89.8 | 80.7|  79.0|   0.02|
> | MagicPIG  |(11,300)  |  90.6    |83.3|  81.9|   0.02|
>
> - Same set of (K, L) can generalize to larger LLMs
>
> |   Models / Config  |  Full | (10,135) |(10, 150) | (9, 110) | (9, 120) |
> |    ------        | -----       | -----|-----  |  -----  |  -----  |
> |    Llama3.1-8B-Instruct   |  86.1      |  83.6 |84.8   | 84.7|   84.7 |
> |    Llama3.1-70B-Instruct   |  89.1      |  86.7 |88.2   | 88.4|  89.1 |

---

> ### Author Response · Authors · 2024-11-22
> **Notes on LSH hyper-parameters (2)**
>
> ## How to select (K, L).
>
> **Finding the optimal (K, L) for high accuracy and efficiency is a long-standing problem in LSH**.  Like the traditional hyperparameter tuning process in machine learning, K and L are configured offline based on data subsets. In LSH, **K is a more sensitive hyperparameter than L**.  A slight change of K can drastically influence the number of retrieved items (i.e., budget) and quality. In MagicPIG, K=8-10 is **manually** determined by ablations on small-scale tasks and found to be effective across various models and tasks. Then, we adjust L to obtain the desired computation cost/budget.
>
> Here,  we present two ablations to demonstrate the selection of K.
>
> Model: Llama-3.1-8K-Instruct; Task: RULER + 16k; Full model accuracy: **94.2**
>
> - $\text{\textcolor{blue}{Exp1: Vary L and fix the computation cost/budget}}$
>
> | K       |  L |  Accuracy |  cost |
> |   ----- | -----| -----     | -----   |
> |  10    |  240|  94.2   | 4%|
> |   9     |  120| 92.8 | 4%|
> |   8     |  65  | 92.3 | 4%|
> |   7     |  35  |   88.5  |  4%|
>
>
> - $\text{\textcolor{blue}{Exp2: Fix L as 120 and vary K (the cost/budget will also vary)}}$
>
> | K       |  L |  ACC |  cost |
> |   ----- | -----| -----     | -----   |
> |  11   |   120| 60.2  |    0.5%|
> |  10    |  120| 87.3 | 1.2%|
> |   9     |  120|92.8 | 4%|
> |   8     |  120| 94.1 | 11%|
> |   7     |  120 | 94.3 | 27%|
>
> If we want the computation cost to be below 5% and L below 200 (to reduce memory overhead in the CPU), then K=8-10 is a reasonable choice. Unlike K, L is not that sensitive. We select L based on the following principle after determining K: for larger K, we can allow the computation cost to be smaller since the sampling is more precise. This is why we choose to use (8, 75), (9, 120), and (10, 150).
>
> It’s worth pointing out that tuning (K, L) is a challenging problem in LSH [1], and we only give a simple example in MagicPIG. More advanced hashing algorithms (such as Cross-polytope [2] or data-dependent ones [3]) can improve the trade-off between memory overhead and accuracy. We leave it as a future direction.
>
> [1] Qin Lv, William Josephson, Zhe Wang, Moses Charikar, and Kai Li. 2017. Intelligent probing for locality sensitive hashing: multi-probe LSH and beyond. Proc. VLDB Endow. 10, 12 (August 2017), 2021–2024. https://doi.org/10.14778/3137765.3137836
>
> [2] Kitaev, Nikita, Łukasz Kaiser, and Anselm Levskaya. "Reformer: The efficient transformer." arXiv preprint arXiv:2001.04451 (2020).
>
> [3] Andoni, Alexandr, and Ilya Razenshteyn. "Optimal data-dependent hashing for approximate near neighbors." Proceedings of the forty-seventh annual ACM symposium on Theory of computing. 2015.

---

### Public Comment · ~Anastasiia_Filippova2 · 2025-02-26

Dear Authors,

Thank you very much for your work! It gave a really solid understanding of the problems behind TopK eviction strategy.

I have a question regarding Algorithm 1: MagicPIG Decoding (page 8). If I understand correctly, you use two types of key-value pairs to compute the final attention: $K_s, V_s$ and $K_T, V_T$. You refer to $K_T, V_T$ as the static KV cache. However, I could not find an explicit reference to static cache in the paper. If I understand correctly, you are referring to the attention sink tokens as well as local tokens. Could you confirm whether this interpretation is correct? If so, could you kindly help me find the ratio of static vs. sampled keys in the paper?

Additionally, you mentioned:
> Sink tokens (the first several tokens) and local tokens are more likely to be sampled according to their high similarity to the query. To further reduce CPU workload, MagicPIG stores these tokens on GPU and does not apply LSH sampling to them.

Correct me if I am wrong: you store sink keys as well as local keys on the GPU and **do not apply hashing to them**.

I could be wrong, but given the above, does this not imply that the estimator you apply is no longer unbiased?

I look forward to your response! This is a really interesting contribution, and I want to be sure that I fully understand the details.

---

> ### Public Comment · ~Zhuoming_Chen1 · 2025-02-26
> **Reply to Comment by Anastasiia Filippova**
>
> Thank you for your interest!
>
> Your understanding of our implementation is correct. However, **the presence of sink tokens does not influence the features of the estimator**.
>
> To clarify this, I will explain in two parts:
>
> **1 Effect of Always Choosing the Attention Sink Tokens**
>
> Selecting attention sink tokens **does not affect** the features of the estimator.  As shown in **Equation (9), Equation (11), and Algorithm 1 (Page 7)**, we adjust the attention score $w_i$ using the **sampling probability** $u_i$.  As long as $u_i$ represents the **actual probability of token $i$ being sampled**, the estimator's features remain preserved.  Thus, we can safely set $u = 1$ for sink tokens without impacting the analysis.
>
> **2 Bias in the MagicPIG Estimator**
>
> The MagicPIG estimator is biased because it relies on the **self-normalized importance sampling estimator** described in **Equation (6)**.  This type of estimator introduces bias, as discussed in [this reference](https://artowen.su.domains/mc/Ch-var-is.pdf).  The bias is **theoretically bounded**, and the estimator maintains a small error and variance when the sampling probability is properly chosen, ensuring strong practical performance.
>
> Let me know if any part needs further clarification!

---

> > ### Public Comment · ~Anastasiia_Filippova2 · 2025-02-28
> >
> > Thank you for your detailed response!
> > I wrongly assumed that the self-normalized importance sampling estimator is unbiased—thank you for pointing me to the reference.
> > Could I also ask if you have tried sampling every token using hashing, without treating sink and local tokens differently?

---

> > > ### Public Comment · ~Zhuoming_Chen1 · 2025-03-01
> > >
> > > I have not gone through a detailed experiment on this.
> > >
> > > Just some experience. Sink and local tokens help MagicPIG perform better. However, the performance is not sensitive to how many local/sink tokens are preserved (I have tried 4/16 for sink tokens and 32/64 for local tokens).

---

### Meta-Review · Area_Chair_4qSx · 2024-12-15

**Metareview:**

This paper studies the optimization of long-context LLM inference and instead of using TopK selection for attention calculation, presents a method based on LSH and importance sampling. Experiments are encouraging.

**Additional Comments On Reviewer Discussion:**

Rebuttal was satisfactory and helped with the assessment.

---

### Decision · Program_Chairs · 2025-01-22

Accept (Spotlight)